

**A large sample analysis of seasonal river flow correlation and its physical**
**drivers**
Theano Iliopoulou[1*], Cristina Aguilar[2], Berit Arheimer[3], María Bermúdez[4], Nejc Bezak[5], Andrea
Ficchì[6], Demetris Koutsoyiannis[1], Juraj Parajka[7], María José Polo[2], Guillaume Thirel[8] and Alberto
Montanari[9]
[(1)] Department of Water Resources and Environmental Engineering, School of Civil Engineering,
National Technical University of Athens, Zographou, 15780, Greece
[(2)] Fluvial dynamics and hydrology research group, Andalusian Institute of Earth System Research,
University of Cordoba, Cordoba, 14071, Spain
[(3)] Swedish Meteorological and Hydrological Institute, 601 76 Norrköping, Sweden
[(4)] Water and Environmental Engineering Group, Department of Civil Engineering, Universidade
da Coruña, 15071 A Coruña, , Spain
[(5)] Faculty of Civil and Geodetic Engineering, University of Ljubljana, Jamova 2, SI-1000
Ljubljana, Slovenia
[(6)] Department of Geography and Environmental Science, University of Reading, Reading, RG6
6AB, United Kingdom; formerly, IRSTEA, Hydrology Research Group (HYCAR), F-92761,
Antony, France
[(7)] Vienna University of Technology, Institute of Hydraulic Engineering and Water Resources
Management, Karlsplatz 13/222, A-1040 Vienna, Austria
[(8)] IRSTEA, Hydrology Research Group (HYCAR), F-92761, Antony, France
[(9)] Department DICAM, University of Bologna, Bologna, 40136, Italy
* *Correspondence to:* Theano Iliopoulou (anyily@central.ntua.gr)



**Abstract**
The geophysical and hydrological processes governing river flow formation exhibit persistence
at several timescales, which may manifest itself with the presence of positive seasonal
correlation of streamflow at several different time lags. We investigate here how persistence
propagates along subsequent seasons and affects low and high flows. We define the High Flow
Season (HFS) and the Low Flow Season (LFS) as the three-month and the one-month periods
which usually exhibit the higher and lower river flows, respectively. A dataset of 224 European
rivers spanning more than 50 years of daily flow data is exploited. We compute the lagged
seasonal correlation between selected river flow signatures, in HFS and LFS, and the average
river flow in the antecedent months. Signatures are peak and average river flow for HFS and
LFS, respectively. We investigate the links between seasonal streamflow correlation and various
physiographic catchment characteristics and hydro-climatic properties. We find persistence to be
more intense for LFS signatures than HFS. To exploit the seasonal correlation in flood frequency
estimation, we fit a bivariate Meta-Gaussian probability distribution to peak HFS flow and
average pre-HFS flow in order to condition the peak flow distribution in the HFS upon river flow
observations in the previous months. The benefit of the suggested methodology is demonstrated
by updating the flood frequency distribution one season in advance in real-world cases. Our
findings suggest that there is a traceable physical basis for river memory which in turn can be
statistically assimilated into flood frequency estimation to reduce uncertainty and improve
predictions for technical purposes.

**Keywords**: flood frequency, seasonal correlation, persistence, real-time flood forecasting, meta-
Gaussian



## 1. Introduction

Recent analyses for the Po River and the Danube River highlighted that catchments may exhibit significant correlation between peak river flows and average flows in the previous months (Aguilar et al., 2017). Such correlation is the result of the behaviours of the physical processes involved in the rainfall-runoff transformation that may induce memory in river flows at several different time scales. The presence of long-term persistence in streamflow has been known for a long time since the pioneering works of Hurst (1951) and has been actively studied ever since (e.g. Koutsoyiannis, 2011; Montanari, 2012; O'Connell et al., 2016 and references therein). While a number of seasonal flow forecasting methods have been explored in the literature (e.g. Bierkens and van Beek, 2009; Dijk et al., 2013), attempts to explicitly exploit streamflow persistence in seasonal forecasting through information from past flows have been in general limited. Koutsoyiannis et al. (2008) proposed a stochastic approach to incorporate persistence of past flows into a prediction methodology for monthly average streamflow and found the method to outperform the historical analogue method (see also Dimitriadis et al., 2016 for theory and applications of the latter) and artificial neural network methods in the case of the Nile River. Similarly, Svensson (2016) assumed that the standardized anomaly of the most recent month will not change during future months to derive monthly flow forecasts for 1–3 months lead time and found the predictive skill to be superior to the analogue approach for 93 UK catchments. A few other studies have included past flow information in prediction schemes along with teleconnections or other climatic indices (Piechota et al., 2001; Chiew et al., 2003; Wang et al., 2009). Recently, it was shown that streamflow persistence, revealed as seasonal correlation, may also be relevant for prediction of extreme events by allowing one to update the flood frequency distribution based on river flow observations in the pre-flood season and reduce its bias and variability (Aguilar et al., 2017). The above previous studies postulated that seasonal streamflow correlation may be due to the persistence of the catchments storage and/or the weather, but no attempt was made to identify the physical drivers.

The present study aims to further inspect seasonal persistence in river flows and its determinants, by referring to a large sample of catchments in 6 European countries (Austria, Sweden, Slovenia, France, Spain



and Italy). We focus on persistence properties of both high and low flows by investigating the following
research question: can floods and droughts be predicted, in probabilistic terms, by exploiting the information
provided by average flows in the previous months? The question is relevant for gaining a better
comprehension of catchment dynamics and planning mitigation strategies for natural hazards. In fact, we also
aim at determining what the physical conditions are, in terms of catchment properties, i.e. geology and
climate, which may induce seasonal persistence in river flow. To reach the latter goal, we identify a set of
descriptors for catchment behaviours and climate, and inspect their impact on correlation magnitude and
therefore predictability.
A few studies have analysed physical drivers of streamflow persistence on annual and deseasonalized
monthly and daily timeseries (Mudelsee, 2007; Hirpa et al., 2010; Gudmundsson et al., 2011; Zhang et al.,
2012; Szolgayova et al., 2014; Markonis et al., 2018) but the topic has been less studied on intra-annual scales
relevant to seasonal forecasting of floods and droughts.
Therefore, we herein follow up previous work by further investigating in a larger sample of catchments
the predictability of high and low flows in probabilistic terms. Additionally, we inspect the physical drivers
of correlation.

## 2.  Methodology

The investigation of the persistence properties of river flows focuses separately on both high and low
discharges and is articulated in the following steps: (a) identification of the high- and low-flow seasons; (b)
correlation assessment between the peak flow in the high flow season (average flow in the low-flow season)
and average flows in the previous months; (c) analysis of the physical drivers for streamflow persistence and
its predictability through a Principal Component Analysis; (d) real-time updating of the flood frequency
distribution for selected case studies with significant seasonal correlation by employing a Meta-Gaussian
approach. The above steps are described in detail in the following sections.



## 2.1 Season Identification

Season identification is performed algorithmically to identify the High Flow Season (HFS) and Low Flow Season (LFS) for each river time series. For the estimation of HFS, we employ an automated method recently proposed by Lee et al. (2015), which identifies the high flow season as the three-month period centred around the month with the maximum number of occurrences of Peaks Over Threshold (POT), with the threshold set to the highest 5 % of the daily flows. To evaluate the selection of HFS, a metric constructed as the Percentage of Annual Maximum Flows (PAMF) captured in the HFS is employed. The PAMFs are classified in subjective categories of "poor" (<40 %), "low" (40–60 %), "medium" (60–80 %) and "high" (>80 %) values, denoting the probability that the identified HFS is the dominant high-flow season in the record. If the identified peak month alone contains 80 % or more of annual maxima flows, a uni-modal regime is assumed and the identification procedure is terminated. In all other cases, the method allows for the search of a second peak month and the identification of a minor HFS but we do not further elaborate on this analysis here because we focus on the major HFS.

The method proposed by Lee et al. (2015) has several advantages that make it suitable for the purpose of this research. Most importantly, it is capable of handling conditions of bi-modality, which is usually a major issue for traditional methods like, e.g., directional statistics (Cunderlik et al., 2004). A potential limitation is the assumption of symmetrical extension of HFS around the peak month, along with the uniform selection of its length (3-month period). The degree of subjectivity in the evaluation of the second HFS is another limitation, which is not relevant here as we focus on the main HFS.

LFS is herein identified as the one-month period with the lowest amount of mean monthly flow. An alternative approach of estimating the relative frequencies of annual minima of monthly flow and selecting the month with the highest frequency as LFS is also considered.

## 2.2 Correlation analysis and physical interpretation through Principal Component Analysis

In the case of HFS, a correlation is sought between the maximum daily flow occurring in the HFS period and the mean flow in the previous months. For LFS, correlation is computed between the mean flow in the LFS





itself and the mean flow in the previous months. Since we are interested in seasonal persistence, we compute
the Pearson's correlation coefficient up to 9-month lag for HFS and 11-month lag for LFS.
An extensive investigation is carried out to identify physical drivers of seasonal streamflow correlation,
in terms of catchment, climatic and geological descriptors.
As catchment descriptors, we consider the basin area ($A$), the Baseflow Index (BI), the mean specific
runoff (SR) and the percentage of basin area covered by lakes (percentage of lakes, PL) and glaciers
(percentage of glaciers, PG) as candidate explanatory variables for streamflow correlation.
The area $A$ (km$^2$) is primarily investigated as it is representative of the scale of the catchment, under
the assumption that in larger basins the impact of the climatological and geophysical processes affecting river
flow becomes more significant and may lead to a magnified seasonal correlation.
BI is considered basing on the assumption that high groundwater storage may be a potential driver of
correlation. BI is calculated from the daily flow series of the rivers following the hydrograph separation
procedure detailed in Gustard et al. (2009). Flow minima are sampled from non-overlapping 5-day blocks of
the daily flow series and turning points in the sequence of minima are sought and identified when the 90 %
value of a certain minimum is smaller or equal to its adjacent values. Subsequently, linear interpolation is
used in between the turning points to obtain the baseflow hydrograph. The baseflow index is obtained as the
ratio of the volume of water beneath the baseflow separation curve versus the total volume of water from the
observed hydrograph, and an average value is computed over all the observed hydrographs for a given
catchment. A low index is indicative of an impermeable catchment with rapid response, whereas a high value
suggests high storage capacity and a stable flow regime.
SR (m$^3$ s$^{-1}$ km$^{-2}$) is computed as the mean daily flow of the river standardized by the size of its basin
area. It may be an important physical driver as it is an indicator of the catchment's wetness. PL (%) and PG
(%) are investigated for the Swedish and Austrian catchments, respectively, as lakes and glaciers are expected
to increase catchment storage thus affecting persistence. Lake coverage data are based on cartography and





available from the Swedish Water Archive (https://www.smhi.se/), while glacier coverage data are estimated
from the CORINE land cover database (https://www.eea.europa.eu/publications/COR0-landcover).
The effect of catchment altitude is also inspected using relief maps from the Shuttle Radar Topography
Mission (SRTM) data (http://srtm.csi.cgiar.org/). The data are available for the whole globe and are sampled
at 3 arch-seconds resolution (approximately 90 meters). Topographic information is available for all
catchments located at latitude lower than 60 degrees north while a 1 km resolution digital elevation model is
available for Austria.
As geological descriptors we consider the percentage of catchment area with the presence of flysch
(percentage of flysch, PF) and karstic formations (percentage of karst, PK) for Austrian and Slovenian
catchments, respectively, for which this type of information is available. A subset of Austrian catchments is
characterised by the dominant presence of flysch, which is known to generate a very fast flow response.
Karstic catchments are also known for having rapid response times and complex behaviour; e.g. initiating
fast preferential groundwater flow and intermittent discharge via karstic springs (Ravbar, 2013; Cervi et al.,
2017). Geological features are expected to be linked to persistence properties also because of geology is the
main control for the baseflow index across the European continent (Kuentz et al. 2017). PK (%) and PF (%)
are estimated from geological maps of Slovenia and Austria, respectively.
As climatic descriptors, the mean annual precipitation $P$ (mm year$^{-1}$) and the mean annual temperature
$T$ (°C) are selected. Data are retrieved from the Worldclim database (http://www.worldclim.org/) at a spatial
resolution of 10 minutes of degree. We also adopt as climatic descriptor the De Martonne index (De
Martonne, 1926), IDM, which is given by $IDM = P/(T + 10)$ , and enables classification of a region into
one of the following  6 climate classes, i.e., arid (IDM ≤ 5), semi-arid (5 < IDM ≤ 10), dry sub-humid (10 <
IDM ≤ 20), wet sub-humid (20 < IDM ≤ 30), humid (30 < IDM ≤ 60) and very humid (IDM ≥ 60).
Additionally, the Köppen-Geiger climatic classification (Kottek et al., 2006) of the rivers is also assessed.
To identify what catchment, physiographic and climatic characteristics may explain river memory we
attempt to regress the seasonal streamflow correlation against the physical descriptors introduced above. We



expect the presence of multi-collinearity among the explaining variables and therefore Principal Component
Analysis (PCA; Pearson, 1901; Hotelling, 1933) was applied to construct uncorrelated explanatory variables.
In essence, PCA is an orthonormal linear transformation of $p$ data variables into a new coordinate system of
$q \leq p$ uncorrelated variables (principal components, PCs) ordered by decreasing degree of variance retained
when the original $p$ variables are projected into them (Jolliffe, 2002). Therefore, the first principal axis
contains the greatest degree of variance in the data, while the second principal axis is the direction which
maximizes the variance among all directions orthogonal to the first principal axis and so on. Specifically, let
$x$ be a random vector with mean $\mu$ and correlation matrix $\Sigma$, then the principal component transformation of
$x$ is obtained as follows:
$$\boldsymbol{y} = \boldsymbol{C}^T \boldsymbol{x}' \tag{1}$$
where $\boldsymbol{y}$ is the transformed vector whose $k$th column is the $k$th principal component ($k = 1, 2..p$), $\boldsymbol{C}$ is the $p \times$
$p$ matrix of the coefficients or loadings for each principal component and $\boldsymbol{x}'$ is the standardized $\boldsymbol{x}$ vector.
Standardization is applied in order to avoid the impact of the different variable units on selecting the direction
of maximum variance, when forming the PCs. The $\boldsymbol{y}$ values are the scores of each observation, i.e. the
transformed values of each observation of the original $p$ variables in the $k$th principal component direction.

PCA has useful descriptive properties of the underlying structure of the data. These properties can be

efficiently visualized in the biplot (Gabriel, 1971), which is the combined plot of the scores of the data for
the first two principal components along with the relative position of the $p$ variables as vectors in the two-
dimensional space. Herein, the distance biplot type (Gower and Hand, 1995), which approximates the
Euclidean distances between the observations, is used. Variable vectors coordinates are obtained by the
coefficients of each variable for the first two principal components. After construction of the PCs, a linear
regression model is explored for the case of HFS and LFS lag-1 correlation.



### 2.3 Technical experiment: Real-time updating of the flood frequency distribution


In order to evaluate the usefulness of the information provided by the one-month-lag seasonal correlation for
HFS, we perform a real-time updating of the flood frequency distribution based on the average river flow in
the previous month. A similar analysis was carried out by Aguilar et al. (2017) for the Po and Danube Rivers.
In detail, a bi-variate meta-Gaussian probability distribution (Kelly and Krzysztofowicz, 1997;
Montanari and Brath, 2004) is fitted between observed peak flow in the HFS, $Q_p$ and the average flow in the
pre-flood season month, $Q_m$. The peak flow is the dependent variable and is extracted as the peak river
discharge observed in the previously identified HFS. The average flow in the month preceding the HFS is
the explanatory variable. In the following, random variables are denoted by underscore and their outcomes
are written in plain form.
The normal quantile transform, NQT (Kelly and Krzysztofowicz, 1997), is used in order to make the
marginal probability distribution of dependent and explanatory variables Gaussian. This is achieved as
follows: a) the sample quantiles $Q$ are sorted in increasing order e.g. $Q_{m_1}, Q_{m_2} \ldots Q_{m_n}$, b) the cumulative
frequency $FQ_{m_i}$ is computed via a Weibull plotting position, and c) the standard normal quantile $NQ_{m_i}$ is
obtained as the inverse of the standard normal distribution for each cumulative frequency, i.e. $G^{-1}(FQ_{m_i})$.
Therefore, all sample quantiles are discretely mapped into the Gaussian domain. To get the inverse
transformation for any normal quantile $NQ_{m_i}$, we connect the points in the above mapping with linear
segments. The extreme segments are extended to allow extrapolation outside the range covered by the
observed sample.
In the Gaussian domain, a bivariate Gaussian distribution is fitted between the random explanatory
variable $\underline{NQ_m}$ and the dependent variable $\underline{NQ_p}$ assuming stationarity and ergodicity of the variables:
$NQ_p(t) = \rho(\underline{NQ_m}, \underline{NQ_p}) NQ_m(t) + N\varepsilon(t)$ (2)
where $\rho(\underline{NQ_m}, \underline{NQ_p})$ is the Pearson's cross correlation coefficient between $\underline{NQ_m}$ and $\underline{NQ_p}$, and $N\varepsilon(t)$ is an
outcome of the stochastic process $\underline{N\varepsilon}$, which is independent, homoscedastic, stochastically independent of
$\underline{NQ_m}$ and normally distributed with zero mean and variance $1-\rho^2(\underline{NQ_m}, \underline{NQ_p})$. Then, the joint bivariate



Gaussian probability distribution function is defined by the mean ($\mu(\underline{NQ_m}) = 0$ and $\mu(\underline{NQ_p}) = 0$), the standard
deviation ($\sigma(\underline{NQ_m}) = 1$ and $\sigma(\underline{NQ_p}) = 1$) of the standardized normalized series, and the Pearson's cross
correlation coefficient between the normalized series, $\rho(\underline{NQ_m,}\ \underline{NQ_p})$. From the Gaussian bivariate probability
properties, it follows that for any observed $NQ_m(t)$ the probability distribution function of $\underline{NQ_p}$ conditioned
on $NQ_m$ is Gaussian, with parameters given by:
$\mu(\underline{NQ_p}) = \rho(\underline{NQ_m,}\ \underline{NQ_p})\ NQ_m$                                   (3)
$\sigma(\underline{NQ_p}) = (1 - \rho^2(\underline{NQ_m,}\ \underline{NQ_p}))^{0.5}$                                   (4)
To derive the probability distribution of $\underline{Q_p}$ conditioned to the observed $Q_m$, we apply the inverse NQT. This
is referred to as the updated probability distribution. We use the Extreme Value Type I distribution for the
peak flows and calculate the differences in the magnitude of estimated maxima for a given return period
between the unconditioned and the updated distribution. The latter is conditioned by the 95% sample quantile
of the observed mean flow in the previous month.
**3.    Data and catchments description**
The dataset includes 224 records spanning more than 50 years of daily river flow data, mostly from non-
regulated streams. A few catchments are impacted by mild regulation. Among the 224 rivers, 108 are located
in Austria, 69 in Sweden, 31 in Slovenia, 13 in France, 2 in Spain and one in Italy. Catchment areas vary
significantly, the largest being the Po River basin in Italy (70 091 km$^2$) and the smaller being the Hålabäck
River basin in Sweden (4.7 km$^2$). The geographical location of the river gauge stations is shown in Fig. 1.
Most of the examined rivers belong to either a warm temperate (C) or a boreal/snow climate (D) with a subset
impacted by polar climatic conditions (E), according to the updated World Map of the Köppen-Geiger climate
classification (Fig. 1) based on gridded temperature and precipitation data for the period 1951-2000 (Kottek
et al., 2006). More specifically, the majority of French, Slovenian and approximately one third of the Swedish
basins belong to the warm temperate Cfb category characterized by precipitation distributed throughout the
year (fully humid) and warm summers. The rest of the Swedish catchments are impacted by a Dfc climatic
type, i.e. a snow climate, fully humid with cool summers. The Austrian catchments belonging to the region



impacted by the European Alps have the most complicated regime due to their topographic variability. At
the lowest altitudes, Cfb is the prevailing regime, but as proximity to the Alps increases, a Dfc regime
dominates and progressively, in the highest altitude basins, the climate becomes a polar tundra type (Et),
characterized primarily by the very low temperatures present. A summary of the river basins under study in
terms of the selected descriptors is also provided in Table 1, showing that the investigated rivers cover a wide
range of catchment area sizes, flow regimes and climatic conditions.
It is interesting to note that some of the above rivers are subject to regulation, which may alter the
persistence properties of river flows. On the one hand, under the assumption that river flow management
does not change in time, the presence of regulation does not preclude the exploitation of correlation for
predicting river flows in probabilistic terms. On the other hand, regulation may affect the analysis of physical
drivers, as it may enhance or reduce persistence in the natural river flow regime. Given that the results that
we herein present are derived from a large sample of catchments, we assume that they are not significantly
affected by the mild regulation that takes place in a few of them.

## 258    4. River memory analysis for the considered case studies

### 259    4.1 Season Identification

Approximately half of the 224 rivers are characterized by at least one high-flow season with medium or
higher significance (PAMF(HFS) ≥ 60 %). Among them, very strong unimodal regimes (PAMF(HFS) ≥ 80
%) are observed in 63 rivers, the majority of which are located in Sweden. For 25% of the rivers, a high-flow
season of low significance is found (PAMF(HFS) between 40–60 %), while for the remaining 25 % the high-
flow distribution looks uniform along the year. Bi-modality regimes are found with low and moderate
significance in rivers located mostly in Austria and Sweden, but we focus here on the major high-flow season,
for which we inspect higher seasonal correlation against previous average flow.
Regarding the LFS identification, the two considered approaches (see Section 2.1) agree for 139 out of
224 stations but the first method, i.e. the one-month period with the lowest amount of mean monthly flow is
selected as being more relevant to the purpose of computing mean flow correlations.



## 4.2 Seasonal correlation

LFS correlation is markedly higher than the corresponding HFS correlation for lags 1–5 and its median remains higher than 0 for more lags (see Fig. 2). For the case of HFS correlation, we focus only on the most significant first lag, for which 73 rivers are found to have correlation significantly higher than 0 at 5 % significance level. In Fig. 3, the autocorrelation of the whole monthly series is compared to the LFS correlation for lag of 1 and 2 months, in order to prove that the seasonal correlation for LFS is significantly higher than its counterpart computed by considering the whole year. The latter is also confirmed by the Kolmogorov-Smirnov test for both LFS lags (corresponding p-values, $p_{lag1} < 2.2 \times 10^{-6}$ and $p_{lag2} < 2.2 \times 10^{-6}$ for the null hypothesis that the LFS correlation coefficients are not higher than the corresponding values for the monthly series autocorrelation; Conover, 1971).

Figure 4 shows the spatial pattern of HFS and LFS streamflow correlations. It is interesting to notice the emergence of spatial clustering in the correlation magnitude, which implies its dependence on different spatially varying physical mechanisms. For example, for HFS, a geographical pattern emerges within France, since the highest correlation coefficients are located in the northern part of the country, which is characterized by oceanic climate and higher baseflow indexes.

## 5. Physical interpretation of correlation

To attribute the detected correlations to physical drivers, we define 6 groups of potential drivers of seasonal correlation magnitude, which are: basin size, flow indexes, presence of lakes and glaciers, catchment elevation, catchment geology, and hydro-climatic forcing. For some of the descriptors the information is available for few countries only.

In what follows, we will use the term "positive (negative) impact on correlation" to imply that an increasing value of the considered descriptor is associated to increasing (decreasing) correlation. For each descriptor, we also report between parentheses the Spearman's rank correlation coefficient $r_s$ (Spearman, 1904) between its value and the considered (LFS or HFS) correlation, and the p-value of the null hypothesis



$r_s = 0$. Spearman's coefficient is adopted in view of its robustness to the presence of outliers and its capability
of capturing monotonic relationships of non-linear type.

### 5.1   Catchment area – Descriptor $A$

Figure 5 shows that there is only a weak positive impact of the catchment area (log-transformed) on
correlation for HFS ($r_s = 0.17$, $p = 0.01$) but a more significant positive one for LFS ($r_s = 0.27$, $p = 5.5 \times 10^{-5}$).
We expected a more pronounced positive impact of the catchment area. The presence of relevant scatter
in the plots also indicates that it is not a key determinant of correlation.

### 5.2   Flow indexes – Descriptors BI and SR

The effect of the BI and SR is shown in Fig. 6. BI (Fig. 6a) appears to be a marked positive driver for LFS
($r_s = 0.6$, $p = 1.8 \times 10^{-23}$) while its effect for HFS is less clear, being weakly positive ($r_s = 0.21$, $p = 0.001$).
As for SR (Fig. 6b), it looks that both LFS and HFS streamflow correlations drop for increasing wetness ($r_s$
$= -0.4$, $p = 4 \times 10^{-10}$ and $r_s = -0.28$, $p = 2.8 \times 10^{-5}$ respectively).

### 5.3   Presence of lakes and glaciers – Descriptors PL and PG

Detailed information on the presence of lakes is available for the 69 Swedish catchments while areal
extension of glaciers is known for the 108 Austrian catchments. Figure 7 shows their impact. The impact of
lake area (Fig. 7a) on correlation for LFS and HFS is not significant but positive ($r_s = 0.10$, $p = 0.399$ and $r_s$
$= 0.12$, $p = 0.347$). The results for glaciers show a positive impact for LFS ($r_s = 0.28$, $p = 0.081$) but negative
for HFS ($r_s = -0.34$, $p = 0.032$). For a meaningful interpretation, these results should be considered in
conjunction with the seasonality of flows for the Austrian catchments. Low flows for the glacier-dominated
catchments are typically occurring in winter months, when glaciers are not contributing to the flow (Parajka
et al., 2009). Thus the observed result for LFS is more likely portraying the impact of low temperature (low
evapotranspiration) and snow accumulation, the latter generally being a slowly varying process. For HFS,
which is typically occurring in the summer months for the considered catchments, flows are mainly
determined by snowmelt which is associated to large variability and reduced persistence (Fig. 7b).



### 5.4  Catchment elevation

The areal coverage of the SRTM data is limited to 60 degrees north and 54 degrees south and therefore, data for the northern part of the Swedish catchments are not available. The rest of the rivers are divided in three regions based on proximity: Region I including the central and eastern part of the Alps and encompassing Austrian, Slovenian and Italian catchments; Region II showing the western part of the Alps and encompassing French and Spanish territory; and Region III including the southern part of Sweden. Figure 8 shows elevation maps along with the location of gauge stations and magnitude of correlations. Elevation seems to enhance LFS correlation which is more evident in the mountainous Region I (Fig. 8). For HFS correlation there is not a prevailing pattern.

In the case of Austrian catchments, a 1 km resolution digital model is also used to extract information on elevation. Figure 9 confirms that there is a positive correlation pattern emerging with elevation for LFS. Based on local climatological information, it can be concluded that the spatial pattern for LFS correlation is reflective of the timing and strength of seasonality of the low flows in Austria, where dry months occur in lowlands during the summer due to increased evapotranspiration and in the mountains during winter (mostly February) due to snow accumulation which is characterised by stronger seasonality compared to the lowlands flow regime (Parajka et al., 2016; see Fig. 1). Concerning HFS in the same region, high flows are significantly impacted by the seasonality of extreme precipitation (Parajka et al., 2010), which is highly variable, with the exception of the rivers where high flows are generated by snowmelt. Therefore, a spatially consistent pattern does not clearly emerge.

### 5.5  Catchment geology – Descriptors PK and PF

Two different geological behaviours are identified which may impact river correlation. We first focus on 21 Slovenian catchments (out of 31) where more than 50 % of the basin area is characterised by the presence of karstic aquifers (percentage of karstic areas PK ≥ 50 %). Figure 10 shows boxplots of the estimated lag-1





correlation coefficient for both HFS and LFS against rivers where PK < 50 %. It is clear that there is a
significant decrease in correlation where karstic areas dominate for both for HFS and LFS.

In a second analysis, we focus on Austrian catchments and investigate the relationship between

correlation and percentage of Flysch coverage, PF. Figure 11 shows that there is not a prevailing pattern in
either case ($r_s$ = 0.13, p = 0.6 for LFS and $r_s$ = –0.19, p = 0.446 for HFS).
### 5.6   Atmospheric forcing – Descriptors *P* and *T*
Figure 12 shows the lag-1 HFS and LFS correlations against estimates of the annual precipitation *P* and
annual mean temperature *T* as well as the De Martonne index IDM. LFS correlation looks more sensitive
than HFS to the above climatic indices, showing a decrease with increasing temperature and also a decrease
with increasing precipitation ($r_s$ = –0.44, p = $3.1 \times 10^{-12}$ for *P* and $r_s$ = –0.57, p = $1.8 \times 10^{-20}$ for *T*).   HFS
correlation      looks      scarcely      sensitive      to      these      variables      ($r_s$      =
–0.17, p = 0.011 for *P* and $r_s$ = 0.08, p = 0.208 for *T*). The IDM (Fig. 12 c) shows a mild decrease of both
LFS ($r_s$ = –0.06, p = 0.368) and HFS correlation with increasing IDM ($r_s$ = –0.17, p = 0.01), while for the
latter there seems to be a clearer trend (lower correlation with higher IDM) in very humid areas (dark blue
points in Fig. 12c).

### 5.7   Physical drivers of high correlation
To gain further insights into the results we select the 20 catchments having the highest streamflow seasonal
correlation coefficients for both HFS and LFS periods in order to investigate their physical characteristics in
relation to the remaining set of rivers. Table 2 summarizes statistics for selected descriptors in order to
identify dominant behaviours. We also compare the number of rivers with distinctive features, i.e. lakes $N_L$
(number of rivers with lakes), glaciers $N_G$ (number of river with glaciers), flysch $N_F$ (number of rivers with
flysch formations) and karst $N_K$ (number of rivers with karstic areas) for the highest correlation group with





those obtained from 1000 randomly sampled 20-cathcment groups from the whole set of considered
catchments to assess whether higher correlation implies distinctive features.

By focusing on HFS, one can notice that the catchments with higher seasonal correlation are

characterised by larger catchment area, higher baseflow index and temperature with respect to the remaining
catchments, and lower specific runoff, precipitation and wetness. Presence of lake, glaciers, karstic and
Flysch areas do not appear significantly effective at a 5 % significance level. More robust considerations can
be drawn for the LFS: higher seasonal correlation is found for larger catchments with higher baseflow index
and lower specific runoff, precipitation and wetness. Decreasing temperature is strongly associated with
higher correlation for the LFS. The presence of lakes plays a significant role both for lag-1 and lag-2
correlations with the latter being also significantly influenced by presence of glaciers.

## 6.   Principal component analysis of the predictors and linear regression

We attempt to fit a linear regression model to relate correlation to physical drivers, in order to support
correlation estimation for ungauged catchments. To avoid the impact of multicollinearity in the regression
while additionally summarize river information, we apply a PCA analysis (see Section 2.2). Although
correlation effects are efficiently dealt with via the PCA, we avoid including highly correlated variables in
the analysis. For example, the De Martonne Index, Precipitation and SR are mutually highly correlated (all
Pearson's cross-correlations are higher than 0.6) and therefore we only consider the SR in the PCA because
it shows a more robust linear relationship with correlation magnitude. We select $A$, BI, SR and $T$ as the
variables to be considered in the PCA. A log transformation is applied on the basin area to reduce impact of
outliers. Table 3 shows the coefficients estimated for each component (the loadings) and the explained
variance. The first principal component is primarily a measure of BI; the second principal component majorly
accounts for $T$ and the third principal component accounts for $A$. There is an evident geographical pattern
emerging by the visualization of countries in the biplot (Fig. 13). Slovenian rivers cluster towards the
direction of increasing SR and $T$, whereas Swedish rivers towards the opposite direction of increasing BI and





decreasing $T$. Austrian rivers, which are the majority, are the most diverse. The first two components together
explain the 70 % of the total variability in the data.

Naturally, the statistical behaviour of the indexes reflects the known local controls for certain rivers.

For example, the observed lowest BI in Slovenia is consistent with the presence of karstic formations for the
majority of the Slovenian rivers, as also is the higher BI in Sweden and Austria, which is related to the
presence of lakes and glaciers in both countries.

In the case of HFS, all the examined linear models (combinations of ln $A$, SR, BI, $P$, $T$, IDM predictors)

failed in explaining the streamflow correlation magnitude. On the contrary, the linear regression model
performs fairly well in explaining the correlation for LFS, with an adjusted $R^2$ value of 0.58 and an F-test
returning a p-value $< 2.2 \times 10^{-16}$. The coefficients for the first three PCs are found significantly different from
zero at a 0.1 % significance level and are included in the regression (see Table 4). The highest coefficient is
obtained for the first PC, which mostly accounts for BI importance. Diagnostic plots from linear regression
for LFS are shown in Fig. 14. There is no clear violation of the homoscedasticity assumption in linear
regression, apart from the presence of a limited number of outliers. There is a certain departure from
normality in the lower tail of the residuals, which relates to the fact that the model performs better in the area
of higher seasonal streamflow correlations and overestimates the lower correlations.

## 7.    Real-time updating of the flood frequency distribution for selected rivers


We apply the technical experiment to two rivers with significant lag-1 streamflow correlation for HFS and
assess the difference in the estimated flood magnitudes. The first river is the Oise River (55 years of daily
flow values) at Sempigny in France with correlation $\rho = 0.54$, which is the 3rd largest lag-1 correlation for
the HFS in our dataset. The second river is the Torsebro River at Helge in Sweden (53 years of daily flow
values). Its lag-1 correlation coefficient for the HFS equals 0.46 which ranks 9th among the rivers. The
Torsebro River has a catchment area of 3665 km$^2$ with lake coverage of 5.4 %, while the Oise River
catchment is slightly larger (4320 km$^2$).



A visual inspection of the residuals plots for both rivers is also performed (Fig. 15a, b) in order to
evaluate the assumption of homoscedasticity of the residuals of the regression model given by Eq. (2). The
residuals do not show any apparent trend and therefore the Gaussian linear model is accepted. Figure 15 (c,
d) shows the conditioned and unconditioned probability distributions of peak flows in the Gaussian domain.
As expected from Eq. (3) and (4), the variance of the updated (conditioned) distribution decreases while the
mean value increases.
After application of the inverse NQT the conditioned peak flows are modelled through the EV1
distribution and compared to the unconditioned (observed) peak flows. The corresponding Gumbel
probability plots for conditioned and unconditioned distributions are shown in Fig. 15 (e, f) for the two rivers.
For the return period of 200 years, the updated distribution shows a 6 % increase in the flood magnitude for
the Oise River (307.7 $m^3$ $s^{-1}$ to 326.44 $m^3$ $s^{-1}$) and a 10 % increase for the Torsebro River (298.07 $m^3$ $s^{-1}$ to
329.22 $m^3$ $s^{-1}$).

## 8.   Discussion and Conclusions

The methodology presented herein aims to progress our physical understanding of seasonal river flow
persistence for the sake of exploiting the related information to improve probabilistic prediction of high and
low flows. The correlation of average flow in the previous months with LFS flow and HFS peak flow was
found to be relevant, with the former prevailing on the latter. This result was expected since the LFS
correlation refers to average flow while the HFS correlation is related to rapidly occurring events. We also
aim to investigate physical drivers for correlation. Therefore, a thorough investigation of the geophysical and
climatological features of the considered catchments was carried out.
We found that increasing basin area and baseflow index are associated with increasing seasonal
streamflow correlation. Within this respect, Mudelsee (2007), Hirpa et al. (2010) and Szolgayova et al.
(2014a) also found positive dependencies of long-term persistence on basin area, Markonis et al. (2018)
found a positive impact too but for larger spatial scales ($> 2 \times 10^4$ $km^2$), while Gudmunsson et al. (2011)
found basin area to have negligible to no impact to the low-frequency components of runoff. Our results



additionally point out that catchment storage induces mild positive correlation, not only for low discharges
which are directly governed by base flow, but also for high flows.
Previous studies also pointed out that correlation increases for groundwater-dominated regimes (Yossef
et al., 2013; Dijk et al., 2013; Svensson, 2016) and slower catchment response times (Bierkens and van Beek,
2009), which concurs with the impact of baseflow index found herein as well as with the observed impact of
fast responding karst areas. The latter findings are also in agreement with our conclusion that correlation
decreases for increasing rapidity of river flow formation, which for instance occurs in the presence of karstic
areas and wet soils, which explains why persistence decreases with high specific runoff; as also confirmed
by other studies (Gudmundsson et al., 2011; Szolgayova et al., 2014).
Other contributions also reported higher streamflow persistence in drier conditions, either relating to
lower specific runoff or mean areal precipitation estimates (Szolgayova et al., 2014; Markonis et al., 2018).
It was postulated that this is due to wet catchments showing increased short-term variability compared to
drier catchments (Szolgayova et al., 2014) and having a faster response to rainfall due to saturated soil. A
similar conclusion has been reached by other previous studies reporting that low humidity catchments are
more sensitive to inter-annual rainfall variability (Harman et al., 2011), therefore leading to enhanced
persistence. Yet, these studies refer to generally humid regions and cannot be extrapolated to more arid
climates. A related conclusion is proposed by Seneviratne et al. (2006) who found the highest soil moisture
memory for intermediate soil wetness. There results do not contrast with our findings, which refer to a wide
range of climatic conditions.
We also confirm the role of lakes in determining higher catchment storage and therefore positive
correlations for the LFS, which has been reported for annual persistence in a few sites (Zhang et al., 2012).
The effect of snow cover for lag-1 LFS correlation is also revealed by the Austrian catchments. The
mountainous rivers, directly affected by the process of snow accumulation, exhibit winter LFS and higher
correlation than the rivers in the lowlands, which are more prone to drying out due to evapotranspiration in
the hotter summer months. The inspection of elevation data confirmed the role of high altitudes in increasing



LFS correlation, which is likely related to storage effects due to snow accumulation and gradual melting. In
this respect, Kuentz et al. (2017) found that topography exerts dominant controls over the flow regime in the
larger European region, controlling the flashiness of flow, and being a particularly important driver for other
low flow signatures too. In fact, topography may affect the flow regime directly, through flow routing, but
also indirectly, because of orographic effects in precipitation and hydroclimatic processes affected by
elevation (e.g. snowmelt and evapotranspiration).

Regarding atmospheric forcing, we find LFS correlation to be negatively correlated to mean areal

temperature and annual precipitation. The former result may be explained considering that increased
evapotranspiration (higher temperature) is expected to dry out LFS flows while snow coverage (lower
temperature) was found to be associated to higher LFS correlation. An apparently different conclusion was
drawn by Szolgayova et al. (2014a) and Gudmundsson et al. (2011), who reported increasing persistence
with increasing mean temperature postulating that snow-dominated flow regimes smooth out interannual
fluctuations. Yet, it should be noted that they refer to interannual variability while we refer here to seasonal
correlation and therefore to shorter time scales, which imply a different dynamic of snow accumulation and
snowmelt; latitude may also play a relevant role in this, since in southern Europe the complete ablation of
snow can occur more than once during the cold season, and sublimation may account for 20–30 % of the
annual snowfall (Herrero and Polo, 2016), decreasing the amount of snowmelt and impacting LFS flows in
the summer season.

Snowmelt mechanisms are found to increase predictive skill during low-flow periods in some other

studies (Bierkens and van Beek, 2009; Mahanama et al., 2011; Dijk et al., 2013). However, in the glacier-
dominated regime of western Alpine and central Austrian catchments this is not expected to be a relevant
driver of higher correlation, since low flow is occurring in the winter months. Yet the mountainous, glacier-
dominated rivers still show increased LFS correlation compared to rivers in the lowlands, which agrees well
with other studies that have found less uncertainty in the rainfall-runoff modelling in this regime owing to





the greater seasonality of the runoff process and the decreased impact of rainfall compared to the rainfall-
dominated regime of the lowlands (e.g Parajka et al., 2016).

Although the considerable uncertainty of areal precipitation estimates should be acknowledged, the

contribution of annual precipitation interestingly complements the negative effect of increasing specific
runoff –which is highly correlated to $P$ estimates– on the correlation magnitude for both LFS and HFS. This
outcome confirms that catchments receiving significant amount of rainfall do show less correlation than drier
regimes.

We conclude that our results are essentially in agreement with the relevant literature and point out the

possibility to exploit river memory within a data assimilation context to reduce uncertainty in the prediction
of future high and low flows. The opportunity of exploiting correlation is not affected by the presence of
regulation, provided the management of river flow does not change in time. Therefore, river memory is an
interesting option to inspect opportunities for improving the prediction of water-related natural hazards.
**Data and Code availability**
The data and code used in this study may be made available to the readers upon request to the corresponding
author.
**Competing interests**
The authors declare that they have no conflict of interest.
**Acknowledgements**
The present work was (partially) developed within the framework of the Panta Rhei Research Initiative of
the International Association of Hydrological Sciences (IAHS). Part of the results were elaborated in the
Switch-On Virtual Water Science Laboratory that was developed in the context of the SWITCH-ON (Sharing
Water-related Information to Tackle Changes in the Hydrosphere – for Operational Needs) project, funded
by the European Union Seventh Framework Programme (FP7/2007-2013) under grant agreement no. 603587.




N. Bezak gratefully acknowledges funding by the Slovenian Research Agency (grants J2-7322 and P2-0180).
M. Bermúdez gratefully acknowledges financial support from the Spanish Regional Government of
Galicia, Postdoctoral Grant Program 2014.

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

## Tables

**Table 1** Summary statistics of the river descriptors. Summary statistics for PL, PG and PF variables are computed only for the subset of catchments with positive values (the total number of catchments is also reported in brackets). PK is used as a categorical variable (PK is either higher or lower than 50 % of catchment area), therefore sample statistics are not computed in this case, but the number of stations with PK ≥ 50 % is reported as 'positive' presence of karst.

| Descriptor (Units) | $A$ (km²) | BI (–) | SR (m³ s⁻¹ km⁻²) | PL (%) | PG (%) | PF (%) | PK (–) | $P$ (mm year⁻¹) | $T$ (°C) | IDM (–) |
|---|---|---|---|---|---|---|---|---|---|---|
| Min | 4.7 | 0.29 | 0.004 | 0.5 | 0.1 | 0.3 | – | 444 | –1.8 | 29.41 |
| Max | 70091 | 0.99 | 0.088 | 19.5 | 56.5 | 100 | – | 1500 | 13.7 | 153.40 |
| Standard deviation | 5904.3 | 0.14 | 0.018 | 4.04 | 15.54 | 32.56 | – | 288.22 | 3.59 | 24.53 |
| Sample size | 224 | 224 | 224 | 69 [69] | 39 [108] | 18 [108] | 21 [31] | 224 | 224 | 224 |



**Table 2** Differences in the mean values between the descriptors of the 20-highest correlation river group for HFS and LFS vs the remaining rivers (204). $N_L$, $N_G$, $N_F$ and $N_K$ columns contain the absolute number of rivers in the higher correlation group with the specific descriptor (presence of lake, glacier, flysch and karst ) with * denoting significance at 5 % significance level (two-sided test) and brackets containing the mean value from the 1000 resampled 20-catchment subsets.

| Descriptor (Units) | $A$ (km$^2$) | BI (–) | SR (m$^3$ s$^{-1}$ km$^{-2}$) | $N_L$ (–) | $N_G$ (–) | $N_F$ (–) | $N_K$ (–) | $P$ (mm year$^{-1}$) | $T$ (°C) | IDM (–) |
|---|---|---|---|---|---|---|---|---|---|---|
| HFS lag1 | +38.7 % | +9.6 % | −36.5 % | 5 [6] | 5 [3] | 1 [2] | 1 [2] | −6.7 % | +11.7 % | −11.3 % |
| LFS lag1 | +358 % | +20.2 % | −47.3 % | 17* [6] | 3 [3] | 0 [2] | 0 [2] | −37.9 % | −80 % | −17.3 % |
| LFS lag2 | +139.7 % | +18.9 % | −40.8 % | 12* [6] | 7* [3] | 0 [2] | 0 [2] | −26.5 % | −64.2 % | −8.8 % |

**Table 3** Loadings of the three Principal Components for ln $A$, SR, BI and $T$. The explained variance of each PC is denoted in parenthesis.

| Predictor variables | PC1 (42.5 %) | PC2 (28.2 %) | PC3 (17 %) | PC4 (12.2 %) |
|---|---|---|---|---|
| ln $A$ | −0.486 | −0.427 | 0.748 | 0.145 |
| SR | 0.48 | 0.483 | 0.652 | −0.332 |
| BI | −0.619 | 0.262 | −0.11 | −0.731 |
| $T$ | 0.385 | −0.718 | −0.04 | −0.577 |

**Table 4** Summary of Linear Regression results for the LFS model. *** indicate a 0.1 % significance level.

| Predictor variables | Estimate | Standard Error | t value | Pr(>|t|) | Adjusted R$^2$ | F-statistic |
|---|---|---|---|---|---|---|
| intercept | 0.659407 | 0.008557 | 77.065 | < 2 ×10$^{-16}$*** | 0.5834 | 104.2 |
| PC1 | −0.110632 | 0.006577 | −16.820 | < 2 ×10$^{-16}$*** | | p-value: |
| PC2 | 0.031761 | 0.008070 | 3.936 | 0.000111*** | | < 2.2 ×10$^{-16}$ |
| PC3 | −0.038999 | 0.010388 | −3.754 | 0.000223*** | | |



**Figures**

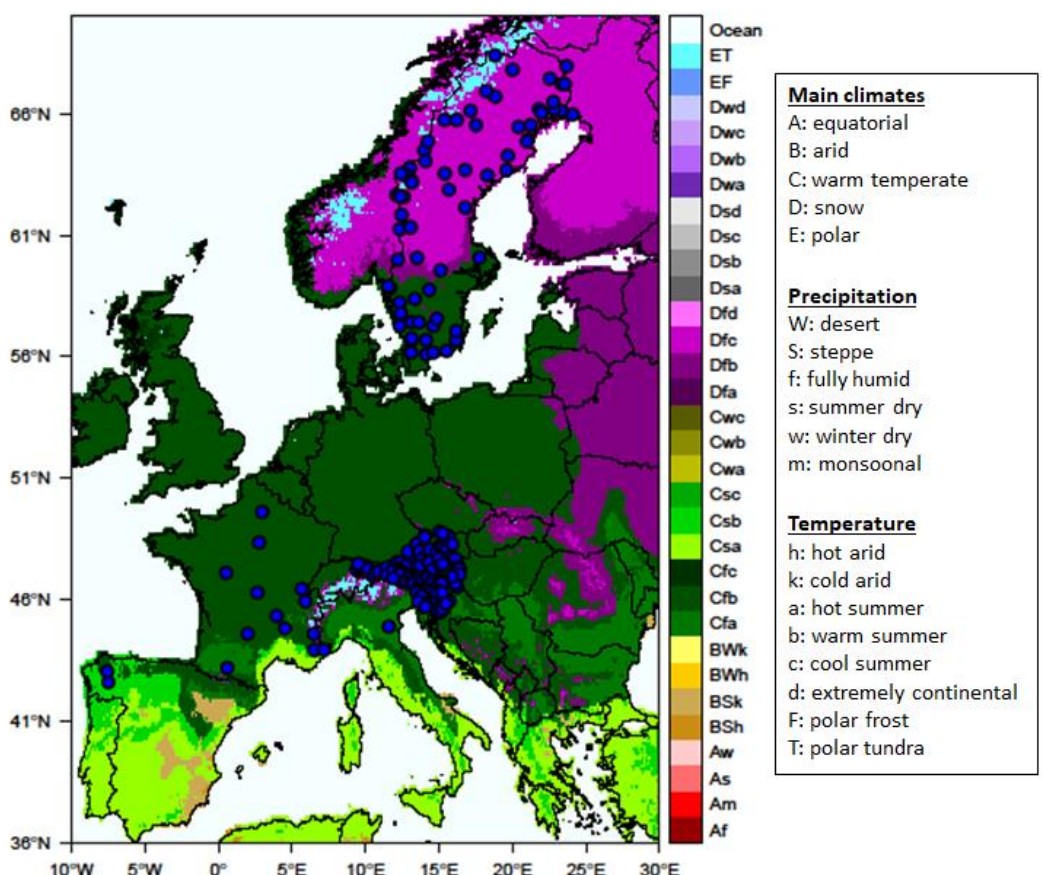


**Figure 1**. Updated Köppen-Geiger climatic map for period 1951–2000 (Kottek et al., 2006) showing the location of
the 224 river gauge stations.





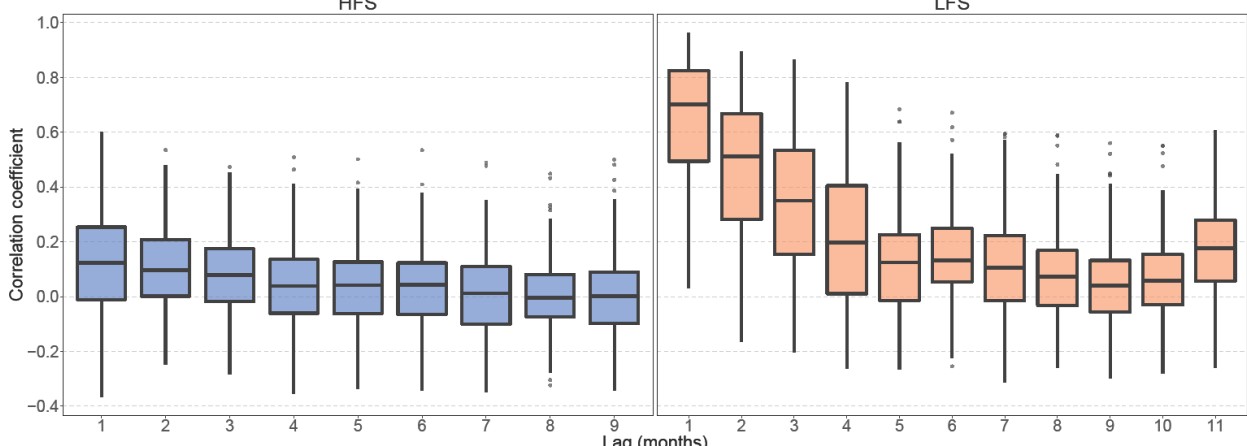

**Figure 2.** Boxplots of seasonal correlation coefficient against lag time for HFS (left panel) and LFS (right panel)
analysis. The lower and upper ends of the box represent the 1st and 3rd quartiles, respectively, and the whiskers extend
to the most extreme value within 1.5 IQR (interquartile range) from the box ends; outliers are plotted as filled circles.

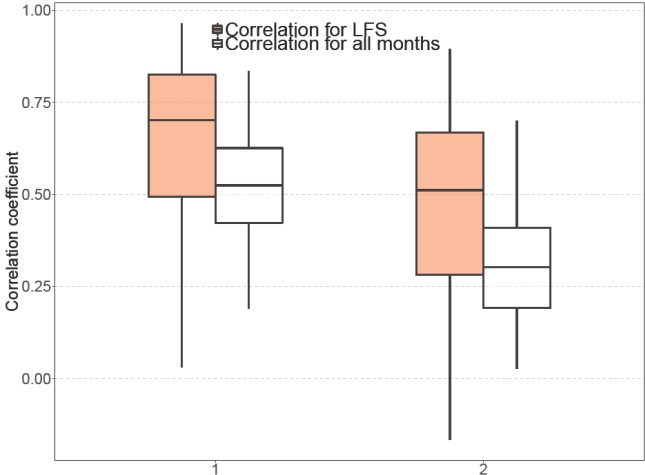

**Figure 3.** Boxplots of lag-1 and lag-2 correlation coefficients for LFS analysis (orange) and the whole monthly series
(white). The lower and upper ends of the box represent the 1st and 3rd quartiles, respectively, and the whiskers extend
to the most extreme value within 1.5 IQR (interquartile range) from the box ends.



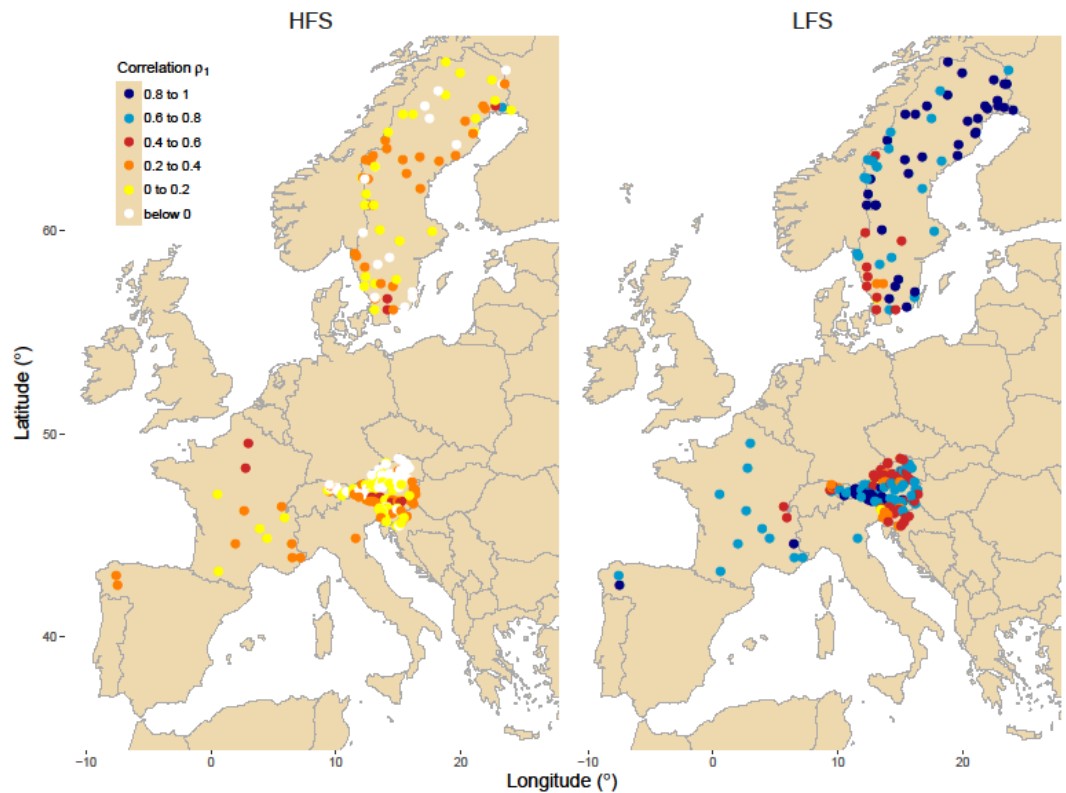

660    .

**Figure 4.** Spatial distribution of the lag-1 correlation coefficients for HFS (left) and LFS (right) analysis. Legend
shows the color assigned to each class of correlation for the data.

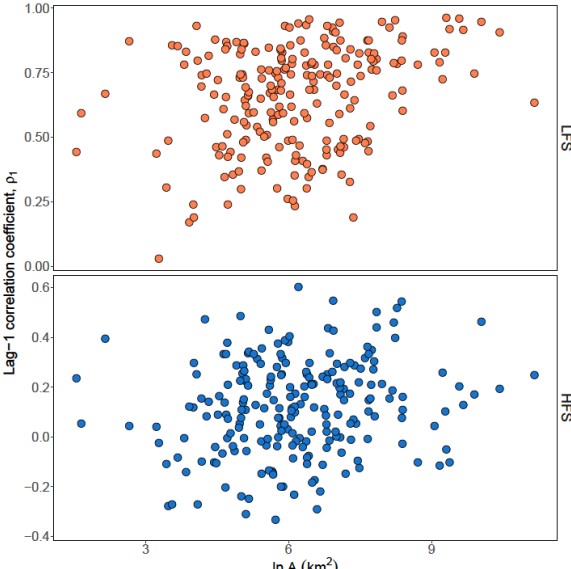

**Figure 5.** Scatterplots of lag-1 HFS (bottom panel) and LFS (top) streamflow correlation versus the natural logarithm
of basin area ln A.





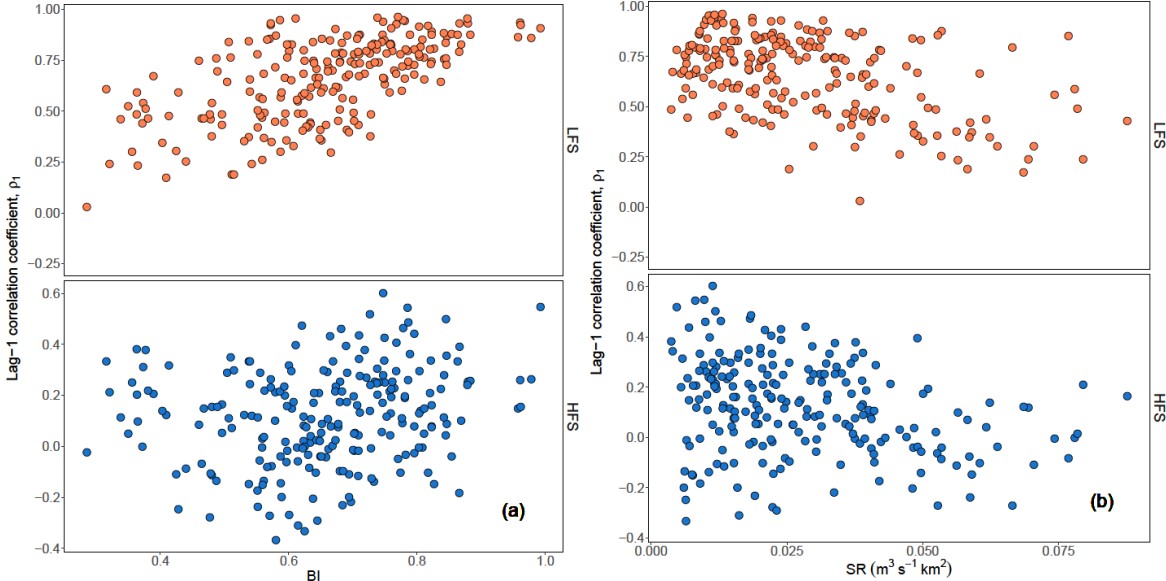

**Figure 6**. Scatterplots of lag-1 HFS (bottom panels) and LFS streamflow correlation (top panels) versus baseflow index BI (a) and specific runoff SR (b).

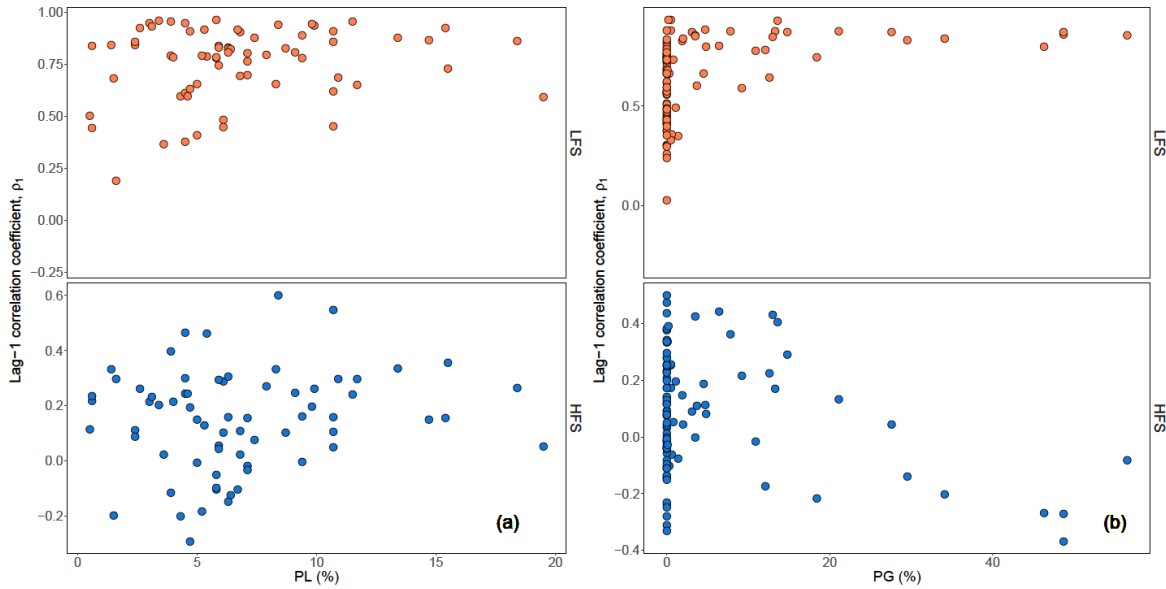

**Figure 7**. Scatterplots of lag-1 HFS (bottom) and LFS (top) streamflow correlations versus percentage of lakes PL of the Swedish catchments (a) and percentage of glaciers PG of the Austrian catchments (b).







**Figure 8**. Relief maps from SRTM elevation data for the HFS and LFS lag-1 correlations of the rivers. Note that
elevation scale is different for each region. Legend shows the colour assigned to each class of correlation for the data.





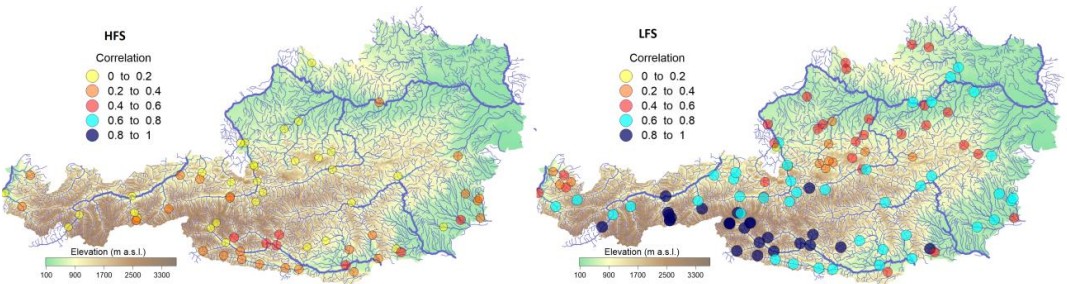

**Figure 9**. Digital elevation model of the Austrian river network depicting the spatial distribution of lag-1 positive
correlation for HFS (left) and lag-1 positive correlation for LFS (right). Legend shows the colour assigned to each
class of correlation for the data.

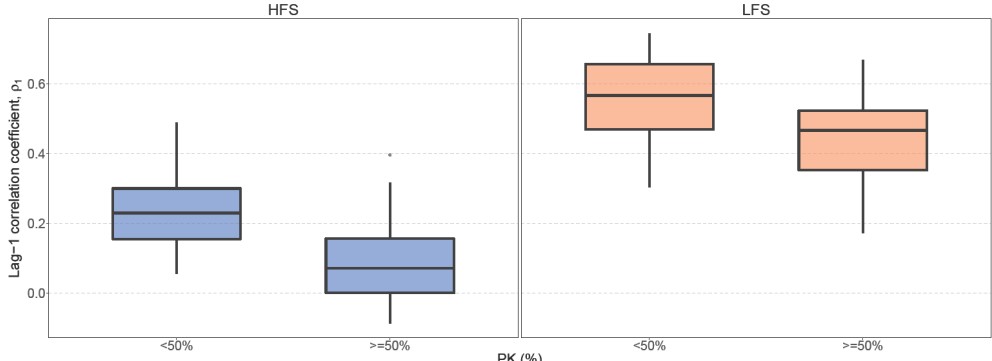

**Figure 10.** Boxplots of lag-1 correlation for Slovenian rivers with more than 50% presence of karstic formations PK
and rivers with no or less presence for HFS analysis (left) and LFS analysis (right). The lower and upper ends of the
box represent the 1st and 3rd quartiles, respectively, and the whiskers extend to the most extreme value within 1.5
IQR (interquartile range) from the box ends.

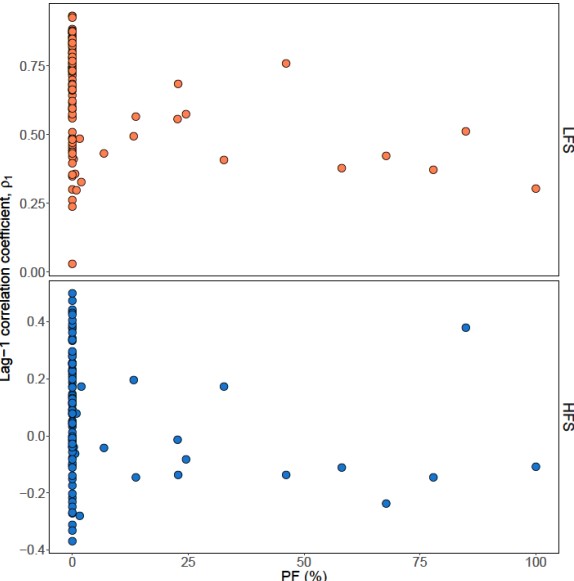

**Figure 11**. Scatterplots of lag-1 correlation vs percentage of flysch area coverage PF for HFS (bottom) and LFS (top)
analysis for the Austrian catchments.




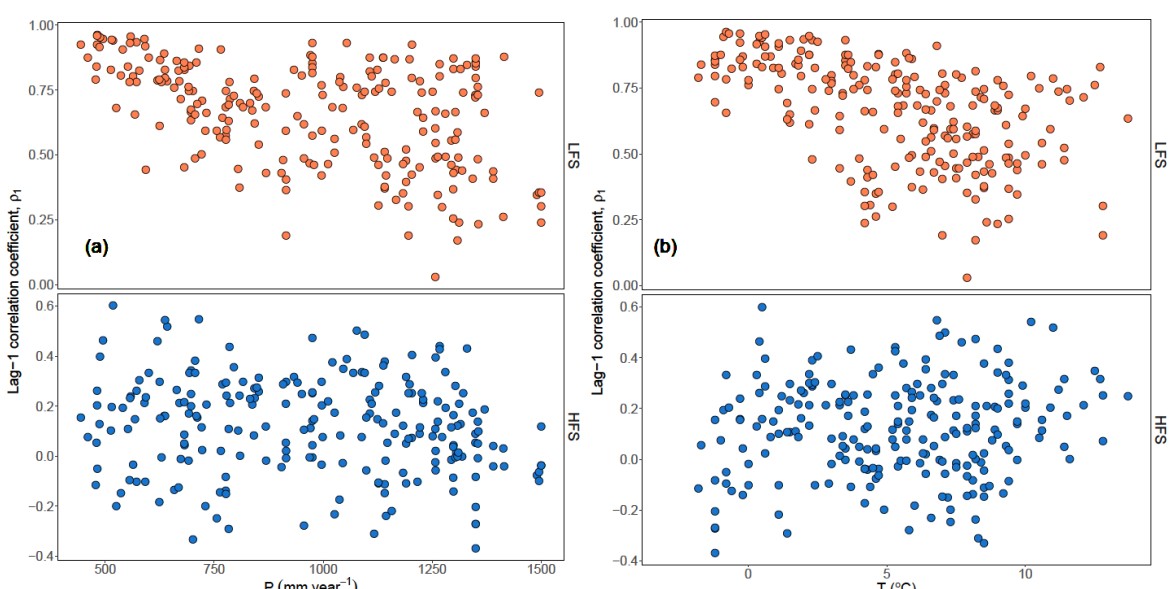


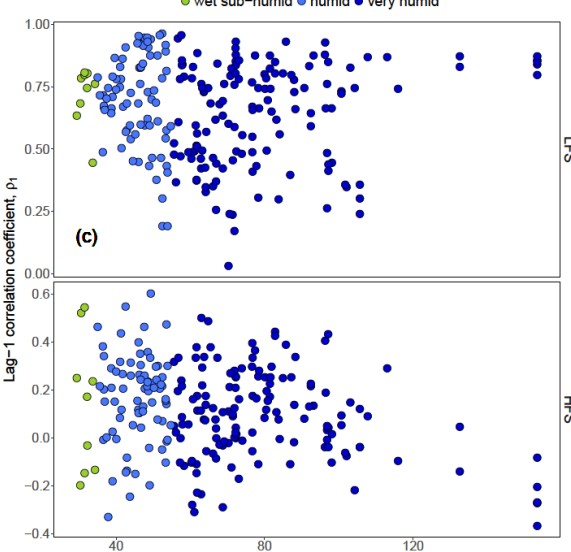

**Figure 12**. Scatterplots of lag-1 HFS and LFS correlation versus annual precipitation P (a), mean annual temperature
T (b), and Index De Martonne IDM (c).





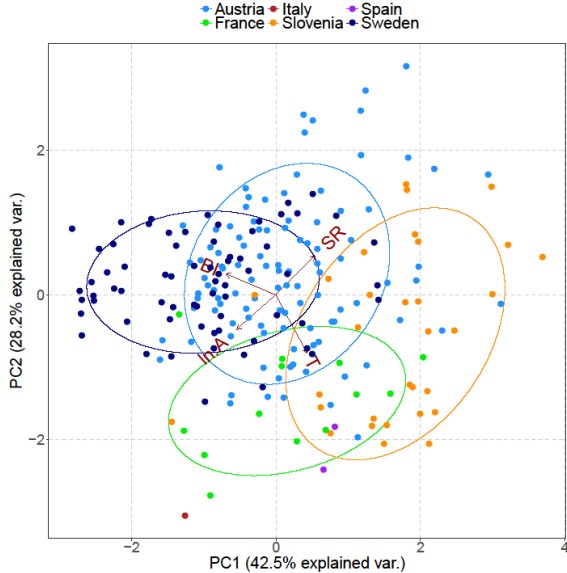

**Figure 13.** Principal component distance biplot showing the principal component scores on the first two principal axes along with the vectors (brown arrows) representing the coefficients of the baseflow index BI, specific runoff SR, natural logarithm of basin area ln $A$ and mean annual temperature $T$ variables when projected on the principal axes. Scores for the rivers are plotted in different colors corresponding to each country of origin and 68% normal probability contour plots are plotted for the countries.

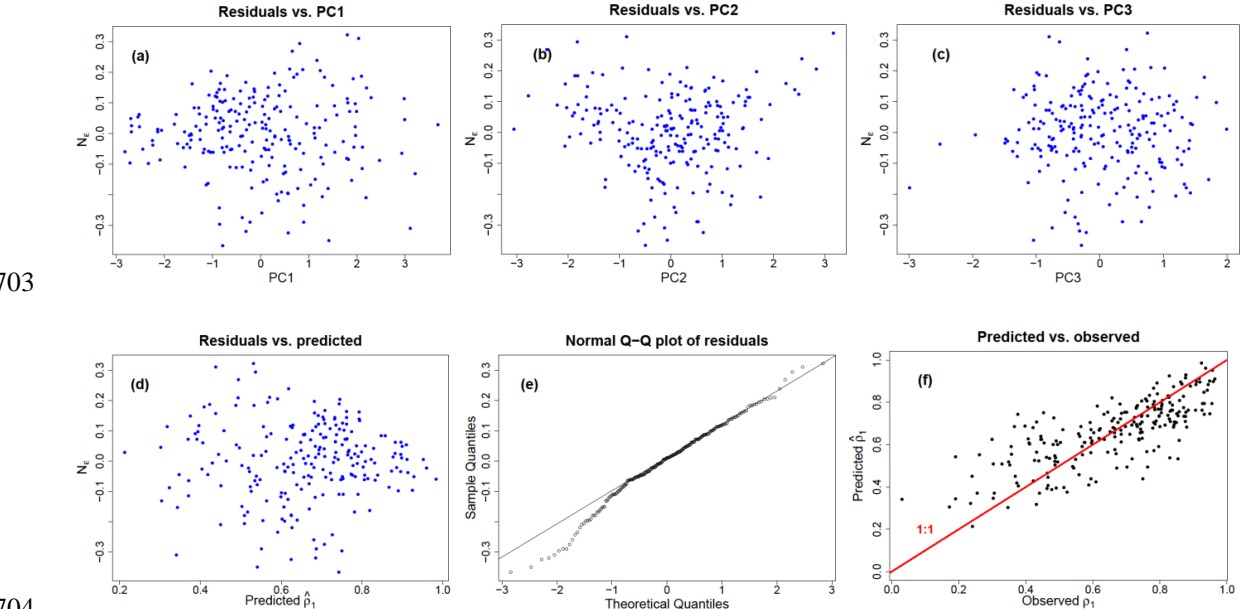

**Figure 14**. Diagnostic plots of linear regression for the LFS model. Residuals versus the first (a), the second (b) and the third principal component (c) and the predicted values (d). Normal Q-Q plot of the residuals (e). Plot of the predicted values from linear regression vs the observed ones; red line is the diagonal line 1:1 (f).





**Figure 15.** Conditioning the flood frequency distribution for the Oise River and the Torsebro River. Plots of the residuals of the linear regression given by Eq. (2) for the Oise River (a) and the Torsebro River (b). Probability distribution of the unconditioned normalized peak flows $NQ_p$ (solid line) and the normalized peak flows $NQ_p$ conditioned to the occurrence of the 95% quantile (dotted line) for the Oise River (c) and the Torsebro River (d). Gumbel probability plots of the return period vs the unconditioned peak flows Qp (black line) and the peak flows Qp conditioned to the occurrence of the 95 % quantile (red line) for the Oise River (e) and the Torsebro River (f).