# Peer review of "A large sample analysis of European rivers on seasonal river flow correlation"

_Hydrology and Earth System Sciences, 2018_

## Referee Comment (RC1) · Anonymous Referee #1 · 27 Apr 2018

SUMMARY:

This paper looks at the lagged seasonal correlations between the average river flow in antecedent months and, on one side, peak flow for the High Flow Season (HFS), and on the other hand, average flow for the Low Flow Season (LFS). It also looks at what are the possible physical drivers that could explain these correlations. The study is carried out using a large sample of European rivers. It also shows a real-case application of the findings to flood frequency estimation.

GENERAL COMMENTS:

The paper is well-written, clear, interesting and attempts more systematically than pre-vious study to attribute the observed correlations to physical drivers. The methods

used are adequate and robust, assumptions are being verified. Overall, it contributes to the advance of science in the field, and my recommendation would therefore be for publication.

However, I have a couple of comments for suggested improvement: 1) My major comment is that, although the whole manuscript looks at both high flows and low flows, and analyses both in detail, the practical example at the end is only for high flows. I think a similar case study for low flows is missing there. If there is a really good reason for only giving an application example for high flows, the motivation for this should be clearly explained.

2) Section 2.2 is too long. It would help readability to have a few sub-sections in here. Suggestion of subsections below (could be different, this is just a suggestion): 2.2.1. Correlation analysis 2.2.2. Analysis of physical drivers a) Drivers (catchment descriptors, geological descriptors, climatic descriptors) b) Principal Component Analysis

MINOR COMMENTS:

Abstract:

line 43: change "in real-world cases" to "in two real-world cases": otherwise it is misleading and it sounds like you've done this to all the 224 catchments.

1. Introduction:

Line 63-66: Note that the persistence method described by Svensson (2016) that you cite here, has been used operationally in the production of the UK Hydrological Outlooks since 2013 (see Prudhomme et al., 2017)

Reference: Christel Prudhomme, Jamie Hannaford, Shaun Harrigan, David Boorman, Jeff Knight, Victoria Bell, Christopher Jackson, Cecilia Svensson, Simon Parry, Nuria Bachiller-Jareno, Helen Davies, Richard Davis, Jonathan Mackay, Andrew McKenzie, Alison Rudd, Katie Smith, John Bloomfield, Rob Ward & Alan Jenkins (2017) Hydrological Outlook UK: an operational streamflow and groundwater level forecasting system

at monthly to seasonal time scales, Hydrological Sciences Journal, 62:16, 2753-2768, DOI: 10.1080/02626667.2017.1395032

2. Methodology

Section 2.2: see comment earlier in general comments regarding splitting this section

Line 127: change "in terms of catchment, climatic and geological descriptors" to "in terms of catchment, geological and climatic descriptors", because that is the order in which you list them later in the text.

Line 128-130: add altitude to the list of catchment descriptors (as you present it after percentages of lakes and glaciers).

Line 139: replace "baseflow index" with "BI"

5. Physical interpretation of correlation

Line 365: typo: replace "20-cathcment" with "20-catchment"

8. Discussion and Conclusions

Line 456: typo: replace "There" with "Their" or "These"

―――――――――――――

---

## Referee Comment (RC2) · Anonymous Referee #2 · 30 Apr 2018

– Recommendation:

This is a very interesting paper, investigating the drivers of seasonal streamflow correlation for both high and low flows, using a wide range of physical drivers including catchment, geological and climatic descriptors. The paper is very well structured, easy to follow, concise and clear throughout with a well explained methodology and clear contribution to the field. Limitations and assumptions are also discussed well. I would recommend this paper for publication subject to minor revisions based on the comments below.

– General Comments:

1. It may be apt to mention that this analysis is for Europe, in the title of the paper

2. I agree with reviewer 1 that the readability of section 2.2 would improve if it were split into subsections

3. It is not clear from the methods or from section 7 why you are doing this technical experiment and what you hope to gain from it. There is a brief explanation of this in the abstract, and it would be beneficial to further describe what the purpose of this experiment is within the manuscript.

4. Again, I agree with reviewer 1 that I was expecting to a case study / technical experiment for low flows as well, and would like to see this included in the revised manuscript as it would certainly be of interest.

5. While I find the discussion to be thorough, with comparison to the literature and interesting points made, the conclusions seem to be very rushed and do not do the paper justice. I would recommend including a separate conclusions section and expanding significantly on this, including for example the wider implications of your work, how the findings could be applied and used, what further work could be done from this, etc. The conclusions imply that all of your results agree with the literature that was already out there, when in fact I believe this paper has done more than this. This is also the first time data assimilation is mentioned so there is no context here. It would also be interesting to further mention section 7 as an example of use.

6. There are a lot of figures included in this manuscript - is it necessary to include all of these, or could some of them be provided in supplementary material for further interest? Some are barely discussed in the paper, for example 15a,b,c,d.

– Minor Comments and Clarifications:

Line 33-34: it should be mentioned that the study covers 6 countries in Europe, the abstract implies that the whole of Europe is included

Line 78: Remove "in fact"

Lines 87-89: This is repetitive of information stated just above

Line 105: "employed" is used a lot in this paragraph - maybe just use "used"?

Line 110-111: Why do you not take into account the minor HFS after identifying it? This could be interesting to discuss; but at least should be justified.

Line 123: Why do you look for correlation with mean flow in the previous months? This is fine, but the reason should be included.

Line 134: basing -> based

Line 155: A very brief explanation of flysch and karstic formations would be helpful for those of us with no geological background.

Line 161: Remove "of" ("because of geology...")

Line 165: What type of data is this?

Line 166: What is this in km (approx.)?

Lines 164-170: You don't mention here how this relates to snow, which is discussed a lot in the results

Line 233: Where is this data from? is it observations? please clarify

Lines 242-243: Please clarify what Cfb and Dfc cliamtic types are

Lines 251: This is indeed interesting, could you expand on which rivers are regulated?

Line 257: Is the regulation really mild; what do you define as mild regulation?

Line 287: indexes -> indices

Line 289: available for "a" few countries only.

Line 204: "it looks that" implies that you are unsure, maybe rephrase this

Lines 349 & 352: again, "looks" implies you are unsure

Line 359: "having" -> "with"
Line 378: summarize -> summarising

Line 378: PCA analysis - analysis is included in this acronym, so reads oddly

Line 385: remove "majorly"

Line 391: indexes -> indices

Line 393: remove "also"

Line 407: add "(see sect. 2.3)" after technical experiment

Line 435: "within this respect" is odd phrasing, consider rephrasing

Line 456: there -> their

Line 473: associated to higher -> associated with higher

Figure 2: Are the boxplots of all the gauging stations? Please clarify in the captions.

Figure 8: Very nice figure, but you have red dots on top of a green map which should ideally be avoided

Figure 9: Again, a very nice figure, but it's very hard to see the yellow dots

---

## Author Comment (AC1) · 29 May 2018

**Reply to the Reviewer#1**

**Key:**

Review comment.

Response.

SUMMARY:

This paper looks at the lagged seasonal correlations between the average river flow in antecedent months and, on one side, peak flow for the High Flow Season (HFS), and on the other hand, average flow for the Low Flow Season (LFS). It also looks at what are the possible physical drivers that could explain these correlations. The study is carried out using a large sample of European rivers. It also shows a real-case application of the findings to flood frequency estimation

GENERAL COMMENTS:

The paper is well-written, clear, interesting and attempts more systematically than previous study to attribute the observed correlations to physical drivers. The methods used are adequate and robust, assumptions are being verified. Overall, it contributes to the advance of science in the field, and my recommendation would therefore be for publication.

We gratefully thank the Reviewer for the very positive evaluation of our work and for recommending publication. We are also thankful for the constructive comments, the corrections and suggestions provided which will certainly help improve the manuscript. These are discussed below.

However, I have a couple of comments for suggested improvement: 1) My major comment is that, although the whole manuscript looks at both high flows and low flows, and analyses both in detail, the practical example at the end is only for high flows. I think a similar case study for low flows is missing there. If there is a really good reason for only giving an application example for high flows, the motivation for this should be clearly explained.

We thank the Reviewer for this comment. Certainly, the application for LFS is also of great importance. We focused on the application for high flows as the relevant methodology for updating the flood frequency distribution using 'river memory' was recently proposed by Aguilar et al. (2017). Therefore, the relevant application for HFS is straightforward. Some modifications are required in order to apply the methodology for the case of predicting average flow in LFS. Following the Reviewer's suggestion, in the revised version, we will present this application too and discuss it.

Section 2.2 is too long. It would help readability to have a few sub-sections in here. Suggestion of subsections below (could be different, this is just a suggestion): 2.2.1. Correlation analysis 2.2.2. Analysis of physical drivers a) Drivers (catchment descriptors, geological descriptors, climatic descriptors) b) Principal Component Analysis

We thank the Reviewer for this suggestion. We agree and we will adopt the proposed subsections.

MINOR COMMENTS:

Abstract:

line 43: change "in real-world cases" to "in two real-world cases": otherwise it is misleading and it sounds like you've done this to all the 224 catchments

1. Introduction:

Line 63-66: Note that the persistence method described by Svensson (2016) that you cite here, has been used operationally in the production of the UK Hydrological Outlook since 2013 (see Prudhomme et al., 2017)

Reference: Christel Prudhomme, Jamie Hannaford, Shaun Harrigan, David Boorman, Jeff Knight, Victoria Bell, Christopher Jackson, Cecilia Svensson, Simon Parry, Nuria Bachiller-Jareno, Helen Davies, Richard Davis, Jonathan Mackay, Andrew McKenzie, Alison Rudd, Katie Smith, John Bloomfield, RobWard & Alan Jenkins (2017) Hydrological Outlook UK: an operational streamflow and groundwater level forecasting system C2 at monthly to seasonal time scales, Hydrological Sciences Journal, 62:16, 2753-2768, DOI: 10.1080/02626667.2017.1395032

2. Methodology

Section 2.2: see comment earlier in general comments regarding splitting this section

Line 127: change "in terms of catchment, climatic and geological descriptors" to "in terms of catchment, geological and climatic descriptors", because that is the order in which you list them later in the text.

Line 128-130: add altitude to the list of catchment descriptors (as you present it after percentages of lakes and glaciers).

Line 139: replace "baseflow index" with "BI"

5. Physical interpretation of correlation
Line 365: typo: replace "20-cathcment" with "20-catchment"

8. Discussion and Conclusions
Line 456: typo: replace "There" with "Their" or "These"

We thank the Reviewer for the above list of minor comments and typos spotted, as well as for bringing to our attention an important application of the persistence method. In the revised version, we will include these suggestions and adopt the corrections provided.

**References**
Aguilar, C., Montanari, A., and Polo, M.-J.: Real-time updating of the flood frequency distribution through data assimilation, Hydrol. Earth Syst. Sci., 21, 3687-3700, https://doi.org/10.5194/hess-21-3687-2017, 2017.

---

## Author Comment (AC2) · 29 May 2018

**Reply to the Reviewer#2**

**Key:**

Review comment.

Response.

Recommendation:

This is a very interesting paper, investigating the drivers of seasonal streamflow correlation for both high and low flows, using a wide range of physical drivers including catchment, geological and climatic descriptors. The paper is very well structured, easy to follow, concise and clear throughout with a well explained methodology and clear contribution to the field. Limitations and assumptions are also discussed well. I would recommend this paper for publication subject to minor revisions based on the comments below.

We are grateful to the Reviewer for providing very positive remarks on the contribution and quality of our work and for recommending publication. We also wish to thank her/him for all the thoughtful suggestions and comments provided which will certainly help improve the manuscript and highlight its contribution. These are discussed below.

General Comments:

1.  It may be apt to mention that this analysis is for Europe, in the title of the paper

Thank you for this suggestion. We will consider a modification of the title in the revised version.

2. I agree with reviewer 1 that the readability of section 2.2 would improve if it were split into subsections.

Thank you for this suggestion. We agree as well and we will adopt Reviewer's 1 suggestion on that.

3. It is not clear from the methods or from section 7 why you are doing this technical experiment and what you hope to gain from it. There is a brief explanation of this in the abstract, and it would be beneficial to further describe what the purpose of this experiment is within the manuscript.

Thank you for the comment. Our intention is to also highlight the practical applicability of this work, besides its importance for improving the physical understanding of river memory. Providing more reliable flood estimates is a fundamental hydrological task and we want to provide a relevant case study showing how the identified correlation explicitly serves such a purpose by exploiting the methodology recently proposed by Aguilar et al. (2017). Following the Reviewer's suggestion, in the revised version, we will elaborate on the purpose of the technical experiment and extend the relevant discussion on the main body of the manuscript as well.

4. Again, I agree with reviewer 1 that I was expecting to a case study / technical experiment for low flows as well, and would like to see this included in the revised manuscript as it would certainly be of interest.

Thank you for the comment. Indeed, this is a very important application too. In the revised version, we will include a relevant case study and discuss its importance as Reviewer 1 has also requested.

5. While I find the discussion to be thorough, with comparison to the literature and interesting points made, the conclusions seem to be very rushed and do not do the paper justice. I would recommend including a separate conclusions section and expanding significantly on this, including for example the wider implications of your work, how the findings could be applied and used, what further work could be done from this, etc. The conclusions imply that all of your results agree with the literature that was already out there, when in fact I believe this paper has done more than this. This is also the first time data assimilation is mentioned so there is no context here. It would also be interesting to further mention section 7 as an example of use.

We sincerely thank the Reviewer for the suggestions on how to improve the conclusions section in order to better convey the research findings of this work. This is an important issue raised and we will consider the suggestions provided thoroughly. We will include a separate conclusions section and discuss areas of practical applicability as well as directions for further research. We also agree with the mentioning section 7 as an example of use and in this section, we will also introduce the data assimilation concept.

6. There are a lot of figures included in this manuscript - is it necessary to include all of these, or could some of them be provided in supplementary material for further interest? Some are barely discussed in the paper, for example 15a,b,c,d.

We thank the Reviewer for the comment. We will consider the possibility to include some of these figures as supplementary material in the revised version.

– Minor Comments and Clarifications:

Line 33-34: it should be mentioned that the study covers 6 countries in Europe, the abstract implies that the whole of Europe is included

Line 78: Remove "in fact"

Lines 87-89: This is repetitive of information stated just above

Line 105: "employed" is used a lot in this paragraph - maybe just use "used"?

Line 110-111: Why do you not take into account the minor HFS after identifying it? This could be interesting to discuss; but at least should be justified.

Line 123: Why do you look for correlation with mean flow in the previous months? This is fine, but the reason should be included.

Line 134: basing -> based

Line 155: A very brief explanation of flysch and karstic formations would be helpful for those of us with no geological background.

Line 161: Remove "of" ("because of geology...")

Line 165: What type of data is this?

Line 166: What is this in km (approx.)?

Lines 164-170: You don't mention here how this relates to snow, which is discussed a lot in the results

Line 233: Where is this data from? is it observations? please clarify

Lines 242-243: Please clarify what Cfb and Dfc cliamtic types are

Lines 251: This is indeed interesting, could you expand on which rivers are regulated?

Line 257: Is the regulation really mild; what do you define as mild regulation?

Line 287: indexes -> indices

Line 289: available for "a" few countries only.

Line 204: "it looks that" implies that you are unsure, maybe rephrase this

Lines 349 & 352: again, "looks" implies you are unsure

Line 359: "having" -> "with"

Line 378: summarize -> summarizing

Line 378: PCA analysis - analysis is included in this acronym, so reads oddly

Line 385: remove "majorly"

Line 391: indexes -> indices

Line 393: remove "also"

Line 407: add "(see sect. 2.3)" after technical experiment

Line 435: "within this respect" is odd phrasing, consider rephrasing

Line 456: there -> their

Line 473: associated to higher -> associated with higher

Figure 2: Are the boxplots of all the gauging stations? Please clarify in the captions.

Figure 8: Very nice figure, but you have red dots on top of a green map which should ideally be avoided

Figure 9: Again, a very nice figure, but it's very hard to see the yellow dots

We thank the Reviewer for the above list of minor comments and suggestions provided as well as for all the errors spotted. In the revised version, we will correct the wording where indicated and provide all the clarifications requested by the Reviewer. We will also improve the readability of Figures 8 and 9 and provide more details on the issue of regulation for the rivers in question.

**References**

Aguilar, C., Montanari, A., and Polo, M.-J.: Real-time updating of the flood frequency distribution through data assimilation, Hydrol. Earth Syst. Sci., 21, 3687-3700, https://doi.org/10.5194/hess-21-3687-2017, 2017.

---

## Author Comment (AC3) · 6 Jul 2018

**Reply to the Reviewers' comments**

**Key:**

Review comment.

Response.

> "Quotation from revised paper."

**Reply to Reviewer #1**

SUMMARY:

This paper looks at the lagged seasonal correlations between the average river flow in antecedent months and, on one side, peak flow for the High Flow Season (HFS), and on the other hand, average flow for the Low Flow Season (LFS). It also looks at what are the possible physical drivers that could explain these correlations. The study is carried out using a large sample of European rivers. It also shows a real-case application of the findings to flood frequency estimation

GENERAL COMMENTS:

The paper is well-written, clear, interesting and attempts more systematically than previous study to attribute the observed correlations to physical drivers. The methods used are adequate and robust, assumptions are being verified. Overall, it contributes to the advance of science in the field, and my recommendation would therefore be for publication.

We gratefully thank the Reviewer for the very positive evaluation of our work and for recommending publication. We are also thankful for the constructive comments, the corrections and suggestions provided which will certainly help improve the manuscript. These are discussed below.

However, I have a couple of comments for suggested improvement: 1) My major comment is that, although the whole manuscript looks at both high flows and low flows, and analyses both in detail, the practical example at the end is only for high flows. I think a similar case study for low flows is missing there. If there is a really good reason for only giving an application example for high flows, the motivation for this should be clearly explained.

We thank the Reviewer for this comment. Certainly, the application for LFS is also of great importance and we agree that it would be a useful addition to the paper. Some modifications are required in order to apply the methodology for the case of updating the distribution of average flow in LFS. Following the Reviewer's suggestion, in the revised version, we will present this application too and discuss it as follows. In order to update the distribution for the average flow in LFS, the already identified average flow in the pre-LFS period will serve as the explanatory variable. A linear model will be adopted in the Gaussian space in the same manner as for the HFS model. For the case study, we will update the low flow distribution

upon the hypothetical occurrence of a mean flow in the pre-LFS month equal to its lower 5% sample quantile. The updated distribution in the non-Gaussian space could be modelled by any adequate distribution exhibiting good fit. Among the Gamma, the Weibull and the lognormal distributions, which are typical candidates for average flows, the lognormal distribution was found to exhibit the best fit for the river in question. The above information along with the equations describing the new linear model for the LFS will be included in Section 2.3 which will be renamed to "Technical experiment: Real-time updating of the frequency distributions for high and low flows". In Section 7, we will include the following plots for the LFS case study alongside the existing plots for the Oise River. In order to maintain the brevity of the manuscript, we will drop the second HFS application for the Torsebro River in the revised version.

[Figure]

[Figure]

[Figure]

**Figure 15.** Conditioning the frequency distribution for low flows for the Oise River. Plots of the residuals of the linear regression given by Eq. (2). Probability distribution of the unconditioned normalized average low flows $NQ_{lm}$ (solid line) and the normalized average low flows $NQ_{lm}$ conditioned on the occurrence of the lower 5% quantile (dotted line). Cumulative distribution function of the unconditioned average low flows $Q_{lm}$ (black line) and the average low flows $Q_{lm}$ conditioned on the occurrence of the lower 5% quantile (red line) and modelled by the lognormal distribution.

Section 2.2 is too long. It would help readability to have a few sub-sections in here. Suggestion of subsections below (could be different, this is just a suggestion): 2.2.1. Correlation analysis 2.2.2. Analysis of physical drivers a) Drivers (catchment descriptors, geological descriptors, climatic descriptors) b) Principal Component Analysis

We thank the Reviewer for this suggestion. We agree and we will adopt the proposed subsections.

MINOR COMMENTS:

Abstract:

line 43: change "in real-world cases" to "in two real-world cases": otherwise it is misleading and it sounds like you've done this to all the 224 catchments

Thanks, we will correct the wording accordingly in the revised version.

1. Introduction:

Line 63-66: Note that the persistence method described by Svensson (2016) that you cite here, has been used operationally in the production of the UK Hydrological Outlook since 2013 (see Prudhomme et al., 2017)

Reference: Christel Prudhomme, Jamie Hannaford, Shaun Harrigan, David Boorman, Jeff Knight, Victoria Bell, Christopher Jackson, Cecilia Svensson, Simon Parry, Nuria Bachiller-Jareno, Helen Davies, Richard Davis, Jonathan Mackay, Andrew McKenzie, Alison Rudd, Katie Smith, John Bloomfield, RobWard & Alan Jenkins (2017) Hydrological Outlook UK: an operational streamflow and groundwater level forecasting system C2 at monthly to seasonal time scales, Hydrological Sciences Journal, 62:16, 2753-2768, DOI: 10.1080/02626667.2017.1395032

We thank the Reviewer for bringing to our attention this important application. We will include this reference in the revised version.

2. Methodology

Section 2.2: see comment earlier in general comments regarding splitting this section

Thanks, we will address this issue as discussed above.

Line 127: change "in terms of catchment, climatic and geological descriptors" to "in terms of catchment, geological and climatic descriptors", because that is the order in which you list them later in the text.

Thanks, we will change this.

Line 128-130: add altitude to the list of catchment descriptors (as you present it after percentages of lakes and glaciers).

Thanks, this will be added.

Line 139: replace "baseflow index" with "BI"

Thanks, we will change it.

5. Physical interpretation of correlation

Line 365: typo: replace "20-cathcment" with "20-catchment"

Thank you for the careful review, we will correct it.

8. Discussion and Conclusions

Line 456: typo: replace "There" with "Their" or "These"

Thanks, we will correct this too.

**Reply to Reviewer #2**

Recommendation:

This is a very interesting paper, investigating the drivers of seasonal streamflow correlation for both high and low flows, using a wide range of physical drivers including catchment, geological and climatic descriptors. The paper is very well structured, easy to follow, concise and clear throughout with a well explained methodology and clear contribution to the field. Limitations and assumptions are also discussed well. I would recommend this paper for publication subject to minor revisions based on the comments below.

We are grateful to the Reviewer for providing very positive remarks on the contribution and quality of our work and for recommending publication. We also wish to thank her/him for all the thoughtful suggestions and comments provided which will certainly help improve the manuscript and highlight its contribution. These are discussed below.

General Comments:

1. It may be apt to mention that this analysis is for Europe, in the title of the paper

Thank you for this suggestion. We will change the title of the paper accordingly.

2. I agree with reviewer 1 that the readability of section 2.2 would improve if it were split into subsections

Thank you for this suggestion. We agree as well and we will adopt Reviewer's 1 suggestion on that.

3. It is not clear from the methods or from section 7 why you are doing this technical experiment and what you hope to gain from it. There is a brief explanation of this in the abstract, and it would be beneficial to further describe what the purpose of this experiment is within the manuscript.

Thank you for the comment. The technical experiment is meant to highlight the practical applicability of the proposed method, besides its importance for improving the physical understanding of river memory. Providing more reliable flood estimates is a fundamental hydrological task and we want to provide a relevant case study showing how the identified correlation explicitly serves such a purpose. Following the Reviewer's suggestion, in the revised version, we will elaborate on the purpose of the technical experiment and extend the relevant discussion on the main body of the manuscript as well. We will also extend the technical experiment to include the low flows distribution updating as well and discuss its practical relevance.

4. Again, I agree with reviewer 1 that I was expecting to a case study / technical experiment for low flows as well, and would like to see this included in the revised manuscript as it would certainly be of interest.

Thank you for the comment. Indeed, this is a very important application too. In the revised version, we will include the relevant case study discussed above.

5. While I find the discussion to be thorough, with comparison to the literature and interesting points made, the conclusions seem to be very rushed and do not do the paper justice. I would recommend including a separate conclusions section and expanding significantly on this, including for example the wider implications of your work, how the findings could be applied and used, what further work could be done from this, etc. The conclusions imply that all of your results agree with the literature that was already out there, when in fact I believe this paper has done more than this. This is also the first time data assimilation is mentioned so there is no context here. It would also be interesting to further mention section 7 as an example of use.

We sincerely thank the Reviewer for the suggestions on how to improve the conclusions in order to better convey the research findings of this work. We will include a separate conclusions section and discuss areas of practical applicability as well as directions for further research. We also agree with mentioning section 7 as an example of use and in this section, we will also introduce the data assimilation concept.

6. There are a lot of figures included in this manuscript - is it necessary to include all of these, or could some of them be provided in supplementary material for further interest? Some are barely discussed in the paper, for example 15a,b,c,d.

We thank the Reviewer for this comment. We will consider the possibility to include some of these figures as supplementary material in the revised version.

– Minor Comments and Clarifications:

Line 33-34: it should be mentioned that the study covers 6 countries in Europe, the abstract implies that the whole of Europe is included

We thank the Reviewer for this remark. We will include this.

Line 78: Remove "in fact"

Thanks, it will be removed.

Lines 87-89: This is repetitive of information stated just above

Thanks, it will be removed.

Line 105: "employed" is used a lot in this paragraph - maybe just use "used"?

Thanks for the suggestion, we will adopt it.

Line 110-111: Why do you not take into account the minor HFS after identifying it? This could be interesting to discuss; but at least should be justified.

We thank the Reviewer for this comment. Actually, we are interested in exploring the river memory for the purpose of predicting high flows and low flows and therefore we are interested in the most extreme seasons. Exploring the memory for the minor HFS may be interesting for reservoir management or water resources management, but in our opinion would not add much for the purpose of analyzing the probability of occurrence of the most relevant flows. Besides, minor high-flow seasons characterized by low or moderate significance were only detected in a few rivers in Austria and Sweden (section 4.1), and therefore, we consider a minor HFS analysis to be more relevant in other regions of the world

where bimodal flood regimes are more prominent, as shown by the analysis of Lee et al. (2015). We will add these considerations in the discussion section of the revised manuscript.

Line 123: Why do you look for correlation with mean flow in the previous months? This is fine, but the reason should be included.

We use the mean flow in the previous month as a robust indicator of the 'storage' in the catchment. The mean flow is more likely to portray the condition of the catchment and its possible change with respect to a higher quantile. The latter correlation is less related to the memory properties of the catchment which are of interest here. We will include the above explanation in the revised version where specified.

Line 134: basing -> based

Thanks, we will correct the wording accordingly.

Line 155: A very brief explanation of flysch and karstic formations would be helpful for those of us with no geological background.

Thanks, we will extend the following phrases giving a brief description of the geology as follows:

"A subset of Austrian catchments is characterized by the dominant presence of flysch, a sequence of sedimentary rocks characterized by low permeability, which is known to generate a very fast flow response."
"Karstic catchments, characterized by the irregular presence of sinkholes and caves, are also known for having rapid response times and complex behaviour; e.g. initiating fast preferential groundwater flow and intermittent discharge via karstic springs (Ravbar, 2013; Cervi et al., 2017)."

Line 161: Remove "of" ("because of geology...")

Thanks, we will remove it.

Line 165: What type of data is this?

'Data' refers to the data described above (mean annual temperature and precipitation), which are gridded. We will clarify this further.

Line 166: What is this in km (approx.)?

10 minutes of degree equal approximately 18.55 km at the equator, i.e. the grid size is approx. 344 km$^2$, but as the latitude increases towards the poles, the longitude distances decrease.

Lines 164-170: You don't mention here how this relates to snow, which is discussed a lot in the results

Thanks, we will add that low mean temperature regimes are associated with snow.

Line 233: Where is this data from? is it observations? please clarify

These are daily streamflow records from gauging stations. These are provided by the institutions mentioned in the authors' affiliations and are available upon request.

Lines 242-243: Please clarify what Cfb and Dfc climatic types are

Thanks, these acronyms are defined in the legend of Figure 1. We will clarify this and reiterate the explanation in the text as well.

Lines 251: This is indeed interesting, could you expand on which rivers are regulated?

Line 257: Is the regulation really mild; what do you define as mild regulation?

We have information for the presence of such regulation for 16 of the Austrian rivers. We used the term 'mild' regulation to describe anthropogenic influences of an intensity that does not majorly alter the flow regimes. These are related to upstream regulation with very low degree of flow attenuation, hydropower operations and flow diversions to and from the basin. Indeed this is a subjective characterization given by the operators of the stations and unarguably the regulation issue requires more investigation. Unfortunately, the data that we have do not have a time reference (start, duration and end of regulation) nor does the regulation have a common starting period for all the rivers in question. A preliminary examination of these rivers did not reveal any consistent patterns worth discussing. However, because regulation is very common in European rivers, although relevant data are generally lacking (Kuentz et al. 2017) and since the possibility of human influences upstream cannot be excluded even in rivers that are formally denoted as nonregulated, we rely on the assumption of stationarity throughout the manuscript. We will include the above explanation of regulation where appropriate in the revised manuscript.

Line 287: indexes -> indices

Line 289: available for "a" few countries only.

Thanks, we will correct the wording accordingly.

Line 204: "it looks that" implies that you are unsure, maybe rephrase this

Lines 349 & 352: again, "looks" implies you are unsure

Thanks for these remarks, we will rephrase.

Line 359: "having" -> "with"

Line 378: summarize -> summarizing

Thanks, we will correct these accordingly.

Line 378: PCA analysis - analysis is included in this acronym, so reads oddly

Indeed, we will rephrase this.

Line 385: remove "majorly"

Line 391: indexes -> indices

Line 393: remove "also"

Thanks, we will adopt the above suggestions.

Line 407: add "(see sect. 2.3)" after technical experiment

Thanks, we will add this.

Line 435: "within this respect" is odd phrasing, consider rephrasing

Thanks, we will rephrase the wording.

Line 456: there -> their

Line 473: associated to higher -> associated with higher

Thanks, we will correct the wording accordingly.

Figure 2: Are the boxplots of all the gauging stations? Please clarify in the captions.

Yes they are. We will add this clarification.

Figure 8: Very nice figure, but you have red dots on top of a green map which should ideally be avoided

Figure 9: Again, a very nice figure, but it's very hard to see the yellow dots

Thank you for pointing this out, indeed this should be avoided. We will change the color of the dots.

Once again, we would like to thank the Reviewers and the Editor for the very constructive assistance.

**References**

Lee, D., Ward, P., and Block, P.: Defining high-flow seasons using temporal streamflow patterns from a global model, Hydrol. Earth Syst. Sci., 19, 4689-4705, https://doi.org/10.5194/hess-19-4689-2015, 2015.

Kuentz, A., Arheimer, B., Hundecha, Y., and Wagener, T.: Understanding hydrologic variability across Europe through catchment classification, Hydrol. Earth Syst. Sci., 21, 2863-2879, https://doi.org/10.5194/hess-21-2863-2017, 2017.

---

## Author Response (AR1)

**Reply to the Reviewers' comments**
* * *
**Key:**

Review comment.

Response.

"Quotation from revised paper."
* * *
**Reply to the Editor**

Dear Authors,

Thank you for your detailed responses to the two referees' reports.

Based on my own reading of the manuscript, I find this is an interesting paper that fits the scope of HESS well.

The two reviews are mostly favourable. However, they also make some valid points about providing context/justification, wider implications (in the discussion), subsections, and moving some of the less important figures to supplementary material (e.g. some of the scatterplots).

I would therefore like to invite you to upload a revised manuscript for further review, incorporating the proposed changes and additions, and making any other modifications where you see fit.

I look forward to receiving the revised manuscript.

With best regards,

Louise Slater

We gratefully thank the Editor Louise Slater for her positive comments and suggestions for improvement as well as for handling the review process. We discuss below how we have taken into account the review comments in the revised version.

**Reply to Reviewer #1**

SUMMARY:

This paper looks at the lagged seasonal correlations between the average river flow in antecedent months and, on one side, peak flow for the High Flow Season (HFS), and on the other hand, average flow for the Low Flow Season (LFS). It also looks at what are the possible physical drivers that could explain these correlations. The study is carried out using a large sample of European rivers. It also shows a real-case application of the findings to flood frequency estimation

GENERAL COMMENTS:

The paper is well-written, clear, interesting and attempts more systematically than previous study to attribute the observed correlations to physical drivers. The methods used are adequate and robust, assumptions are being verified. Overall, it contributes to the advance of science in the field, and my recommendation would therefore be for publication.

We gratefully thank the Reviewer for the very positive evaluation of our work and for recommending publication. We are also thankful for the constructive comments, the corrections and suggestions provided which have certainly helped improve the manuscript. These are discussed below.

However, I have a couple of comments for suggested improvement: 1) My major comment is that, although the whole manuscript looks at both high flows and low flows, and analyses both in detail, the practical example at the end is only for high flows. I think a similar case study for low flows is missing there. If there is a really good reason for only giving an application example for high flows, the motivation for this should be clearly explained.

We thank the Reviewer for this comment. Certainly, the application for LFS is also of great importance and we agree that it would be a useful addition to the paper. Some modifications are required in order to apply the methodology for the case of updating the distribution of average flow in LFS. Following the Reviewer's suggestion, in the revised version, we have presented this application too and discussed it as follows. In order to update the distribution for the average flow in LFS, the already identified average flow in the pre-LFS period serves as the explanatory variable. A linear model is adopted in the Gaussian space in the same manner as for the HFS model. For the case study, we update the low flow distribution upon the hypothetical occurrence of a mean flow in the pre-LFS month equal to its lower 5% sample quantile. The updated distribution in the non-Gaussian space could be modelled by any adequate distribution exhibiting good fit. Among the Gamma, the Weibull and the lognormal distributions, which are typical candidates for average flows, the lognormal distribution was found to exhibit the best fit for the river in question. The above information along with the equations describing the new linear model for the LFS is included in Section 2.3 which is renamed to "Technical experiment: Real-time updating of the frequency distributions for high and low flows". In Section 7, we now include the following plots for the LFS case study alongside the existing plots for the Oise River. In order to maintain the brevity of the manuscript, we have dropped the second HFS application for the Torsebro River. We have also added discussion on the relevance of the chosen river for preparing for extremes occurring in high and low flow periods (Lines 450-457 of the revised manuscript).

[Figure]

[Figure]

**Figure 13.** Conditioning the frequency distributions for high and low flows for the Oise River. Plots of the residuals of the linear regression given by Eq. (2) for HFS model (a) and the LFS (b). Probability distribution of the unconditioned normalized peak flows $NQ_P$ (solid line) and the normalized peak flows $NQ_p$ conditioned to the occurrence of the 95% quantile (dotted line) for the HFS (c) and probability distribution of the unconditioned normalized low flows $NQ_L$ (solid line) and the normalized low flows $NQ_L$ conditioned to the occurrence of the 5% quantile (dotted line) for the LFS (d). Gumbel probability plots of the return period vs the unconditioned peak flows $Q_P$ (black line) and the peak flows $Q_P$ modelled by the EV1 distribution and conditioned to the occurrence of the 95% quantile (red line) for the HFS (e) and cumulative distribution function of the unconditioned low flows $Q_L$ (black line) and the low flows $Q_L$ modelled by the lognormal distribution and conditioned to the occurrence of the 5% quantile (red line) for the LFS (f).

Section 2.2 is too long. It would help readability to have a few sub-sections in here. Suggestion of subsections below (could be different, this is just a suggestion): 2.2.1. Correlation analysis 2.2.2. Analysis of physical drivers a) Drivers (catchment descriptors, geological descriptors, climatic descriptors) b) Principal Component Analysis

We thank the Reviewer for this suggestion. We agree and we have adopted the proposed subsections.

MINOR COMMENTS:

Abstract:

line 43: change "in real-world cases" to "in two real-world cases": otherwise it is misleading and it sounds like you've done this to all the 224 catchments

Thanks, we have corrected the wording to "a real-world case".

1. Introduction:

Line 63-66: Note that the persistence method described by Svensson (2016) that you cite here, has been used operationally in the production of the UK Hydrological Outlook since 2013 (see Prudhomme et al., 2017)

Reference: Christel Prudhomme, Jamie Hannaford, Shaun Harrigan, David Boorman, Jeff Knight, Victoria Bell, Christopher Jackson, Cecilia Svensson, Simon Parry, Nuria Bachiller-Jareno, Helen Davies, Richard Davis, Jonathan Mackay, Andrew McKenzie, Alison Rudd, Katie Smith, John Bloomfield, RobWard & Alan Jenkins (2017) Hydrological Outlook UK: an operational streamflow and groundwater level forecasting system C2 at monthly to seasonal time scales, Hydrological Sciences Journal, 62:16, 2753-2768, DOI: 10.1080/02626667.2017.1395032

We thank the Reviewer for bringing to our attention this important application. We have included a short mention to the operational use of the method in Lines 67-69:
> "The abovementioned persistence approach has also been used operationally in the production of seasonal streamflow forecasts in the UK since 2013, within the framework of the Hydrological Outlook UK (Prudhomme et al. 2017)"

2. Methodology

Section 2.2: see comment earlier in general comments regarding splitting this section

Thanks, we have addressed this issue as discussed above.

Line 127: change "in terms of catchment, climatic and geological descriptors" to "in terms of catchment, geological and climatic descriptors", because that is the order in which you list them later in the text.

Thanks, we have changed this.

Line 128-130: add altitude to the list of catchment descriptors (as you present it after percentages of lakes and glaciers).

Thanks, this is added.

Line 139: replace "baseflow index" with "BI"

Thanks, we have changed it.

5. Physical interpretation of correlation

Line 365: typo: replace "20-cathcment" with "20-catchment"

Thank you for the careful review, we have corrected it.

8. Discussion and Conclusions

Line 456: typo: replace "There" with "Their" or "These"

Thanks, we have corrected this too.

**Reply to Reviewer #2**

Recommendation:

This is a very interesting paper, investigating the drivers of seasonal streamflow correlation for both high and low flows, using a wide range of physical drivers including catchment, geological and climatic descriptors. The paper is very well structured, easy to follow, concise and clear throughout with a well explained methodology and clear contribution to the field. Limitations and assumptions are also discussed well. I would recommend this paper for publication subject to minor revisions based on the comments below.

We are grateful to the Reviewer for providing very positive remarks on the contribution and quality of our work and for recommending publication. We also wish to thank her/him for all the thoughtful suggestions and comments provided which have certainly helped improve the manuscript and highlight its contribution. These are discussed below.

General Comments:

1. It may be apt to mention that this analysis is for Europe, in the title of the paper

Thank you for this suggestion. We have changed the title of the paper to "A large sample analysis of European rivers on seasonal river flow correlation and its physical drivers".

2. I agree with reviewer 1 that the readability of section 2.2 would improve if it were split into subsections

Thank you for this suggestion. We agree as well and we have adopted Reviewer's 1 suggestion on that.

3. It is not clear from the methods or from section 7 why you are doing this technical experiment and what you hope to gain from it. There is a brief explanation of this in the abstract, and it would be beneficial to further describe what the purpose of this experiment is within the manuscript.

Thank you for the comment. The technical experiment is meant to highlight the practical applicability of the proposed method, besides its importance for improving the physical understanding of river memory. Providing more reliable flood estimates is a fundamental hydrological task and we want to provide a relevant case study showing how the identified correlation explicitly serves such a purpose. Following the Reviewer's suggestion, in the revised version, we have elaborated on the purpose of the technical experiment and extended the relevant discussion in the introduction (Lines 91-94) and in the conclusions (Lines 563-573) as well. We have also extended the technical experiment itself (Section 7) to include the low flows distribution updating as well and discussed its practical relevance specifically for the river in question (Lines 450-457):

> "The Oise River (55 years of daily flow values) at Sempigny in France has a basin area of 4320 km$^2$ and its gauging station at Sempigny is part of the French national real-time monitoring system (https://www.vigicrues.gouv.fr/), which is in place to monitor and forecast floods in the main French rivers. The selected river has a high technical relevance since it experiences both types of extremes with large impacts. For instance, a severe drought event in 2005 led to water restrictions impacting agriculture and water uses in the region (Willsher, 2005), while the river originated an inundation during the 1993 flood events in northern and central France, which was one of the most catastrophic flood-related disasters in Europe in the period 1950-2005 (Barreldo, 2007).

4. Again, I agree with reviewer 1 that I was expecting to a case study / technical experiment for low flows as well, and would like to see this included in the revised manuscript as it would certainly be of interest.

Thank you for the comment. Indeed, this is a very important application too. In the revised version, we have included the relevant case study discussed above.

5. While I find the discussion to be thorough, with comparison to the literature and interesting points made, the conclusions seem to be very rushed and do not do the paper justice. I would recommend including a separate conclusions section and expanding significantly on this, including for example the wider implications of your work, how the findings could be applied and used, what further work could be done from this, etc. The conclusions imply that all of your results agree with the literature that was already out there, when in fact I believe this paper has done more than this. This is also the first time data assimilation is mentioned so there is no context here. It would also be interesting to further mention section 7 as an example of use.

We sincerely thank the Reviewer for the suggestions on how to improve the conclusions in order to better convey the research findings of this work. We have included a separate conclusions section (Section 9, "Conclusions and outlook") and discussed areas of practical applicability as well as directions for further research. We have also mentioned section 7 as an example of use and elaborated on other possible probabilistic models. Additionally, we introduced the data assimilation concept in Section 2.3 (Lines 215-217) as follows:

"In principle, this is a data assimilation approach, since real-time information, i.e. observations of the average river flow, is used in order to update a probabilistic model and inform the forecast of the flow signature of the upcoming season."

6. There are a lot of figures included in this manuscript - is it necessary to include all of these, or could some of them be provided in supplementary material for further interest? Some are barely discussed in the paper, for example 15a,b,c,d.

We thank the Reviewer for this comment. We have added Figures 7 and 11 as supplementary material in the revised version.

– Minor Comments and Clarifications:

Line 33-34: it should be mentioned that the study covers 6 countries in Europe, the abstract implies that the whole of Europe is included

We thank the Reviewer for this remark. We have included this.

Line 78: Remove "in fact"

Thanks, we have removed it.

Lines 87-89: This is repetitive of information stated just above

Thanks, it has been removed.

Line 105: "employed" is used a lot in this paragraph - maybe just use "used"?

Thanks for the suggestion, we have adopted it.

Line 110-111: Why do you not take into account the minor HFS after identifying it? This could be interesting to discuss; but at least should be justified.

We thank the Reviewer for this comment. Actually, we are interested in exploring the river memory for the purpose of predicting high flows and low flows and therefore we are interested in the most extreme seasons. Exploring the memory for the minor HFS may be interesting for reservoir management or water resources management, but in our opinion would not add much for the purpose of analyzing the probability of occurrence of the most relevant flows. Besides, minor high-flow seasons characterized by low or moderate significance were only detected in a few rivers in Austria and Sweden (section 4.1), and therefore, we consider a minor HFS analysis to be more relevant in other regions of the world where bimodal flood regimes are more prominent, as shown by the analysis of Lee et al. (2015). We have add these considerations in Lines 301-305 of the revised manuscript:

> "Bi-modality regimes are found with low and moderate significance in rivers located mostly in Austria and Sweden, but we focus here on the major high-flow season, as we are interested in the most extreme events. A minor HFS analysis would be perhaps relevant in other regions of the world where bimodal flood regimes are more prominent, as suggested by the analysis of Lee et al. (2015)."

Line 123: Why do you look for correlation with mean flow in the previous months? This is fine, but the reason should be included.

We use the mean flow in the previous month as a robust indicator of the 'storage' in the catchment. The mean flow is more likely to portray the condition of the catchment and its possible change with respect to a higher quantile. The latter correlation is less related to the memory properties of the catchment which are of interest here. We have include the following explanation in the revised version in Lines 130-132.

> "We use the mean flow in the previous month as a robust proxy of 'storage' in the catchment expected to reflect the state of the catchment, i.e. wetter/drier than usual."

Line 134: basing -> based

Thanks, we have corrected the wording accordingly.

Line 155: A very brief explanation of flysch and karstic formations would be helpful for those of us with no geological background.

Thanks, we have extend the following phrases giving a brief description of the geology as follows:

"A subset of Austrian catchments is characterized by the dominant presence of flysch, a sequence of sedimentary rocks characterized by low permeability, which is known to generate a very fast flow response."

"Karstic catchments, characterized by the irregular presence of sinkholes and caves, are also known for having rapid response times and complex behaviour; e.g. initiating fast preferential groundwater flow and intermittent discharge via karstic springs (Ravbar, 2013; Cervi et al., 2017)."

Line 161: Remove "of" ("because of geology...")

Thanks, we have removed it.

Line 165: What type of data is this?

'Data' refers to the data described above (mean annual temperature and precipitation), which are gridded. We have added the following clarification in Lines 176-177:

"As climatic descriptors, the mean annual precipitation $P$ (mm year$^{-1}$) and the mean annual temperature $T$ (°C) are selected. Corresponding gridded data are retrieved…"

Line 166: What is this in km (approx.)?

minutes of degree equal approximately 18.55 km at the equator, i.e. the grid size is approx. 344 km$^2$, but as the latitude increases towards the poles, the longitude distances decrease. We have included this in parenthesis.

Lines 164-170: You don't mention here how this relates to snow, which is discussed a lot in the results

Thanks, we have add that low mean temperature regimes are associated with snow (Line 179).

Line 233: Where is this data from? is it observations? please clarify

These are daily streamflow records from gauging stations. These are provided by the institutions mentioned in the authors' affiliations and are available upon request. We have added the following phrase (Lines 264-265):

"The dataset includes 224 records spanning more than 50 years of daily river flow observations from gauging stations, mostly from non-regulated streams."

Lines 242-243: Please clarify what Cfb and Dfc climatic types are

Thanks, these acronyms are defined in the legend of Figure 1. We have clarified this and reiterated the explanation in the text as well.

Lines 251: This is indeed interesting, could you expand on which rivers are regulated?

Line 257: Is the regulation really mild; what do you define as mild regulation?

We have information for the presence of such regulation for 16 of the Austrian rivers. We used the term 'mild' regulation to describe anthropogenic influences of an intensity that does not majorly alter the flow regimes. These are related to upstream regulation with very low degree of flow attenuation, hydropower operations and flow diversions to and from the basin. Indeed this is a subjective characterization given by the operators of the stations and unarguably the regulation issue requires more investigation. Unfortunately, the data that we have do not have a time reference (start, duration and end of regulation) nor does the regulation have a common starting period for all the rivers in question. A preliminary examination of these rivers did not reveal any consistent patterns worth discussing. However, because regulation is very common in European rivers, although relevant data are generally lacking (Kuentz et al. 2017) and since the possibility of human influences upstream cannot be excluded even in rivers that are formally denoted as nonregulated, we rely on the assumption of stationarity throughout the manuscript. We have included the above explanation of regulation in Lines 284-293.

> "It is relevant to note that 16 of the Austrian rivers are subject to regulation, which may alter the persistence properties of river flows. This relates to generally 'mild' forms of regulation, i.e. upstream regulation with very low degree of flow attenuation, hydropower operations and flow diversions to and from the basin. A preliminary examination of these rivers did not reveal any significant change during time of the flow regime. The presence of regulation does not preclude the exploitation of correlation for predicting river flows in probabilistic terms, but may affect the analysis of physical drivers, as it may enhance or reduce persistence in the natural river flow regime. Given that detailed information is generally lacking on the impact of regulation (Kuentz et al. 2017), we assume stationarity of the river flows for all the catchments herein considered and additionally, assume that river management does not significantly affect the identification of the physical drivers."

Line 287: indexes -> indices

Line 289: available for "a" few countries only.

Thanks, we have corrected the wording accordingly.

Line 204: "it looks that" implies that you are unsure, maybe rephrase this

Lines 349 & 352: again, "looks" implies you are unsure

Thanks for these remarks, we have rephrased.

Line 359: "having" -> "with"

Line 378: summarize -> summarizing

Thanks, we have corrected these accordingly.

Line 378: PCA analysis - analysis is included in this acronym, so reads oddly

Indeed, we have now removed "analysis".

Line 385: remove "majorly"

Line 391: indexes -> indices

Line 393: remove "also"

Thanks, we have adopted the above suggestions.

Line 407: add "(see sect. 2.3)" after technical experiment

Thanks, we have added this.

Line 435: "within this respect" is odd phrasing, consider rephrasing

Thanks, we have rephrased the wording.

Line 456: there -> their

Line 473: associated to higher -> associated with higher

Thanks, we have corrected the wording accordingly.

Figure 2: Are the boxplots of all the gauging stations? Please clarify in the captions.

Yes they are. We have added this clarification in the caption.

Figure 8: Very nice figure, but you have red dots on top of a green map which should ideally be avoided

Figure 9: Again, a very nice figure, but it's very hard to see the yellow dots

Thank you for pointing this out, indeed this should be avoided. We have changed the color of the dots.

Once again, we would like to thank the Reviewers and the Editor for the very constructive assistance.

**References**

[revised manuscript text omitted]

(2)

where $\rho(NQ_m, NQ_{fs})$ is the Pearson's cross correlation coefficient between $NQ_m$ and $NQ_{fs}$, $h$ is the selected correlation lag with $h = 1$ in the present application, and $N\varepsilon(t)$ is an outcome of the stochastic process $N\varepsilon$, which is independent, homoscedastic, stochastically independent of $NQ_m$ and normally distributed with zero mean and variance $1-\rho^2(NQ_m, NQ_{fs})$. Then, the joint bivariate Gaussian probability distribution function is defined by the mean ($\mu(NQ_m) = 0$ and $\mu(NQ_{fs}) = 0$), the standard deviation ($\sigma(NQ_m) = 1$ and $\sigma(NQ_{fs}) = 1$) of the standardized normalized series, and the Pearson's cross correlation coefficient between the normalized series, $\rho(NQ_m, NQ_{fs})$. From the Gaussian bivariate probability properties, it follows that for any observed $NQ_m(t - h)$ the probability distribution function of $NQ_{fs}(t)$ conditioned on $NQ_m$ is Gaussian, with parameters given by:

$$\mu(NQ_{fs}(t)) = \rho(NQ_m, NQ_{fs})\, NQ_m(t - h)$$

(3)

[revised manuscript text omitted]

---

## Author Response (AR2)

**Reply to the Reviewers' Comments**

We would like to thank the Editor and the Referees for reviewing the revised version of our paper. We acknowledge that one reviewer suggests publication of our manuscript in the present form, while the second reviewer suggests rejection.

While we appreciate the constructive approach adopted by the Editor, and felt very comfortable with the first round of the review process, we would like to kindly point out that we do feel uncomfortable with the second review round. The reason is that one **new** review (the report by Reviewer #2 in the second round) is not open and therefore the review process is not transparent as it should be (according to the journal's policy).

The problem is originated by the fact that a **new** reviewer was involved in the second review round (who did not respond to the invitation to review the paper in the first round). Therefore, his/her report in the second round is actually a **first round review**, which should be open and therefore publicly available. In fact, according to our understanding of the journal's policy, the second review round aims to assess whether the criticism expressed in the public review was successfully addressed or not. New criticism by a new reviewer should not be expressed. In fact, the email we received after the publication of our paper in HESS-D reads as (cut and pasted text is reported between asterisks, with relevant text in red):

\*\*\*\*

----- Mensaje reenviado de editorial@copernicus.org -----Fecha: Tue, 3 Apr 2018 09:11:07 +0200 (CEST) De: editorial@copernicus.org Asunto: hess-2018-134 (author) - manuscript available for public review and discussion

You are receiving the following email copy due to your co-authorship of the manuscript hess-2018-134. The original message was sent to the contact author defined upon manuscript registration. Please contact us in case of any discrepancies with regard to the manuscript.

Dear Theano Iliopoulou,

We are pleased to inform you that your following manuscript has been posted as a discussion paper in HESSD, the scientific discussion forum of HESS:

**OMISSIS**

As soon as the open discussion phase is over, no more referee comments or short comments will be accepted. During the following final response phase, however, you will have the opportunity to post final author comments. Before submitting a revised version of your manuscript for publication in HESS, you are obliged to have answered all referee comments and relevant short comments in one or more author comments in the discussion forum of your paper.

**OMISSIS**

\*\*\*\*

We decided to submit our paper to HESS because we appreciate the transparency of the open review process and we appreciate the opportunity of the public reply to the reviewer comments. If a **new reviewer**, and therefore a **new report**, is involved in the second round, the distinguishing feature of the open review system vanishes. This is particularly relevant in this case as we do not agree with the **new** concerns that were raised by the reviewer in the second review round and therefore we would like to have the opportunity to publicly reply. We are confident that the Editor

will recognize that our reply below is providing interesting arguments that, therefore, deserve to be known by the community.

Furthermore, we would like to stress that the audience would never know the real reason why the paper was not published if the review is not made open and our paper is finally rejected. This would be in contrast with the essential feature of the open review, namely, transparency.

Finally, we would like to point out that the second-round policy that was adopted here may stimulate reviewers to skip the first review round to avoid open publication of their report, therefore annihilating the benefit of submitting papers to HESS.

Therefore, we kindly ask the Editor that the review report by Reviewer #2, alongside with the present document which reports our replies, is published in the open discussion of the first review round. We believe that publication of reviews is important to keep the editorial process of HESS fully transparent.

Here below we reply to the concerns of Reviewer #2.

**Reply to Reviewer #2**

In the following, the comments of the Reviewer are copied in italic.

We first reply to the **general comment** of the Reviewer that reads as:

In the author's own words, the results are often 'expected' and the discussion section mostly 'confirms' previous work and understanding of what is controlling catchment streamflow.

First, we feel it is necessary to clarify that the results were mostly not "expected". In fact, we use the term "expected" several times in the paper to highlight the conjectures that led us to design our experiment. In fact (lines and text refer to the revised version of the paper from the first round reviews):

- at line 130 we write "We use the mean flow in the previous month as a robust proxy of 'storage' in the catchment that is expected to reflect the state of the catchment, i.e., wetter/drier than usual";
- at line 156 we write "...as lakes and glaciers are expected to increase catchment storage thus affecting persistence";
- at line 173 we write "Geological features are expected to be linked to persistence properties...";
- at line 188 we write "We expect the presence of multi-collinearity among the explaining variables and therefore Principal Component....";
- at line 520 we write "The former result may be explained considering that increased evapotranspiration (higher temperature) is expected to dry out LFS flows....";
- at line 532 we write "However, in the glacier dominated regime of western Alpine and central Austrian catchments this is not expected to be [equivalent to "expected not to be"] a relevant driver of higher correlation".

In other cases, we highlight that the results were **not** "expected". In fact:

- at line 338 we write "...indicates that it is not a key determinant of correlation";
- at line 347 we write "The impact of lake area (Fig. S1a) on correlation for LFS and HFS is not significant but positive...";

- at line 374 we write "Therefore, a spatially consistent pattern does not clearly emerge...";
- at line 384 we write "Figure S2 in the Supplement shows that there is not a prevailing pattern in either case...";
- at line 408 we write "Presence of lake, glaciers, karstic and Flysch areas do not appear significantly effective at a 5 % significance level.";

Finally, only in some cases we indeed point out that the results confirmed our expectation and/or the outcome of previous studies. For instance:

- at line 464 we write "As expected from Eq. (3) and (4), the variance of the updated (conditioned) distribution decreases while the mean value increases.";
- at line 479 we write "This result was expected since the LFS correlation refers to average flow while the HFS correlation is related to rapidly occurring events." Please note that this sentence is relevant to our reply to comment #2 below.

Indeed, when we found a potentially interesting result we tried to provide physically based reasoning, and/or review of the previous literature, to give further support to our findings, namely, to provide evidence that they are not merely due to "noise". This is what a rigorous scientific approach requires, rather than a sign of "conspiracy" (see the unfortunate wording that is used by the Reviewer in his/her comment #2 that is copied below).

Actually, we do see the fact that results confirm our previous conjectures as a positive outcome. When a deductive approach is used, the scientist first elaborates a conceptual reasoning to explain what is observed. In this case, we did observe that **peak flows in the high flow season** (HFS) are often preceded by high **mean flows in the previous month**. Therefore, in a previous work we decided to explore the correlation between the two random variables above (highlighted in bold) for two rivers only. The results confirmed our expectation. Therefore, the present contribution aims to (1) extend the analysis to the low flow in the LFS, (2) extend the analysis to several other rivers, and (3) explore the physical drivers of river memory.

About the latter issue, we of course needed to select physically based metrics to explain correlation. We conjectured what physical properties (metrics) may determine correlation and therefore elaborated an expectation. We therefore designed the experiment precisely with the aim to confirm our conjecture. The Reviewer seems to imply that confirmation of conjectures (expectations) makes the results meaningless. We regret to report that we disagree. Rather, confirmation of expectations means that the experiment is well designed.

1. The following aspects of the methodology are unclear: For the HFS, the max daily discharge in the 3-month HFS is chosen. Is this value distinct from the max yearly discharge? In most cases, I suspect not. If the max discharge is in the second month of the HFS, does the lag-1 represent the correlation with the previous months mean discharge (also technically in the 3 month HFS), or with the last month before the onset of the HFS? If it is the latter, then the analysis is no longer technically a lag-1 analysis, and the study could be a big mix of lag-1, lag-2 and even lag-3 analyses that are all confused as representing a lag-1 value. Of course, it could be argued that you wish to be outside the HFS season for the correlation analysis, but what is the value of having inconsistent time periods in your lag analysis, especially given how sensitive the correlation will be to changing lag lengths? Moreover, what if a single catchment has max discharge always moving between the first and third month of the HFS over all the years of record? This will have a large impact on the 'lag-1' correlation even before any hydrological interpretations are involved. Some clarification on the mechanics of this analysis would really help.

Strictly speaking, the Reviewer is right, but a mixed lag is not infrequent in hydrological analysis. As an example when we examine daily maximum discharges of consecutive years, we usually speak about the average time lag which is one year, but in fact this is mixed and varies between 1 day (if the max values were observed in 1 Jan of one year and 31 Dec of the previous year) to 730 days (if the max values were observed in 31 Dec of one year and 1 Jan of the previous year). In our view the important thing is to clarify the terminology and the methodology, and consistently define the related random variables. This does not necessarily require that data are sampled at regular time step or that the time distance of consecutive high flow events is constant.

In our case, we rigorously define in the paper the random variables which we consider. For instance, for the HFS they are:

- Peak flow in the high flow season (with arbitrary but rigorously identified length);
- Mean flow in the previous months.

We also clearly define that we denote with lag-1 the correlation between the peak flow in the HFS and the average flow in the previous month, before the onset of HFS. In the same way we define lag-2 correlation and so on. We regret to report that we do not agree with the criticism of the Reviewer and therefore did not make any major change to the manuscript in this respect. However, we have added this further clarification in the revised manuscript: "In the case of HFS, a correlation is sought between the maximum daily flow occurring in the HFS period and the mean flow in the previous months, before the onset of HFS." (Line 128-129).

2. The authors do not consider how the design of their study may have conspired to control the reported results before referring to a myriad of hydrological explanations. The core issue is one of signal vs noise. The LFS lag analysis uses a correlation between mean values that are by definition weighted by the central tendency of the data being considered, whereas the HFS uses a correlation between a max value and a mean, which is by design a far noisier signal, and hence displays little to no correlation with other variable throughout the study. Can the authors image a scenario where this would not be an expected result?

We fully agree with the Reviewer that correlation between monthly data is expected to be higher with respect to correlation between local variables like peak flow. This is precisely the reason why the correlation that we found between peak flow in high flow season and average flow in previous month is a relevant (and not expected) result. It implies timely predictability of the probability distribution of peak flows, which is a relevant finding.

As for the low flows, we demonstrated that the correlation that we found is higher than the correlation computed for the whole set of monthly data. This means that focusing on the specific correlation of the monthly flow for the LSF season and the monthly flows of the previous months again allows us to improve predictability of low flows. Again, this is not an expected result. In both cases, we found that there is a specific signal that emerges above other signals and noise. Please note: it's not just a question of signal versus noise, which highlights an oversimplified view of the inherent processes. It's a matter of recognizing a specific signal – namely, correlation between previous monthly flows and LFS low flows and HFS peak flow – over other signals (monthly correlation, for instance, for the LFS) and random components.

Turning to the physical explanation that we sought, we do not see the reason why the fact that the results were expected would downgrade their value (please see our reply to the Reviewer's general comment above). Therefore we rebut the statement that we designed the experiment by "conspiring" (a very unfortunate term, as we already remarked) in order to obtain expected results. Again, the experiment was based on our preliminary conjectures that are in turn based on conceptual and

physical reasoning. The fact that the results confirm expectation is a confirmation that the experiment was well designed. For what reason should we investigate possible physical explanations that are not expected to be sound?

Still, we would like to point out once again that many of the explanatory metrics we investigated turned out to be not effective on the correlation, such as, for instance, the presence of lakes and glaciers for the HFS, catchment elevation, flysch areas and so on. Therefore, not all of our results were expected.

To mitigate the concern of the Reviewer, we changed the wording throughout the manuscript to avoid many repetitions of the term "expected". We also made changes in the Discussion section to better highlight the purpose of the analysis and underline more some of the most important and less expected results. The relevant sentences of the revised manuscript (copied at the end of the present report) read:

- At line 482: "We also aim to investigate physical drivers for correlation and quantify their relative impact on correlation magnitude."
- At line 486: "We found that increasing basin area and baseflow index are associated with increasing seasonal streamflow correlation, yet the latter has a stronger impact."
- At line 492: "Our results additionally point out that catchment storage induces mild positive correlation, not only for low discharges which are directly governed by base flow, but also for high flows, which is less anticipated."
- At line 509: "In fact, our finding that increased wetness has a negative impact on seasonal memory of both high and low flows, extends the above results to the seasonal scale and interestingly, to both types of extremes."
- At line 513: "We also confirm the role of lakes in determining higher catchment storage and therefore positive correlations for the LFS, which has only been reported for annual persistence in a few sites (Zhang et al., 2012)."
- 3. Related to point 2, the authors use a suite of metrics, many of which (P, SR, BFI) have a natural correlation with HFS and LFS since they are either derived from the same data or help generate it. The HFS analysis produces such a noisy signal that no result can be found, and this is hardly a surprising result (as mentioned above). LFS is not as noisy, and so displays better correlations. The heart of the paper is then to say that the correlations are better with hydrological processes that will also natural reduced the noise, e.g. higher groundwater flow subsidies and snowmelt, and worse correlations with processes and drivers that have increased noise. Again, can the authors think of a situation where this would not be an expected result?

First, we believe there is a misunderstanding here. We did find that correlation for the HFS season is relevant and helpful to improve predictability. Please see Section 4.2 and 7. Therefore there is indeed a signal that we discovered over what the Reviewer terms "noise". Furthermore, we demonstrated that such correlation is explained by catchment area, precipitation and catchment storage in general. Therefore we regret to report that we cannot agree with the statement that "*The heart of the paper is then to say that the correlations are better with hydrological processes that will also natural reduced the noise*" (sic). The heart of our paper is stated in the last sentence of the abstract: "Our findings suggest that there is a traceable physical basis for river memory which in turn can be statistically assimilated into high- and low-flow frequency estimation to reduce uncertainty and improve predictions for technical purposes."

Furthermore, we do not understand the criticism by the Reviewer "*natural correlation with HFS and LFS since they are either derived from the same data or help generate it*". For instance, we analyzed the correlation between rainfall and river flow. Would the fact that rainfall generates river flow make the analysis of their correlation meaningless? We regret to say that we cannot agree.

4. Given these factors, it is unclear what processes or understanding can be revealed by such an analysis, since the study is producing most of the results by design, rather than by hydrological insight. In this sense the analysis in this manuscript obscures the actual hydrology, for example if you just plot actual baseflow on the maps in Figures 7 and 8 a clearer pattern of the hydrological controls on low flows would be revealed (or indeed baseflow against elevation, as documented by a lot of previous work). Surely, the LFS lag analysis only obscures these key hydrological drivers rather than making them clearer or easier to understand? I think this is also clearly shown in the discussion section, which is highly speculative about general processes and mostly confirms the results of previous workers rather than adding new understanding.

We are glad that the Reviewer recognizes the value of previous studies that analyzed the correlation between baseflow and low flow, even if both baseflow and low flows are "*derived from the same data*". We believe our contribution provides relevant new findings such as:

- We confirmed **by referring for the first time to a large set of basins** that the peak flow probability distribution and the low flow probability distribution can be usefully updated in real time one or more months in advance through data assimilation.
- The physical drivers of predictability of low flows and high flows are **quantitatively identified for the first time for the chosen variables** (please note that graphical depictions may provide a more immediately clear representation, as we all know, but do not allow a quantitative assessment unless a quantitative relationship is provided, as we did).

5. I don't see the value or utility of section 7, it is incredibly short and not at all mentioned in the discussion section, therefore its completely unclear what we have learnt from this exercise, or in what context it's results should be considered. This asymmetry is considerable given it has more length devoted to describing the methodology than anything else in the paper (section 2.3). However, after reading it a couple of times I found this to be the most interesting part of the paper, since it asks an interesting question about how you would expect HFS or LFS to change based on obtaining the new average discharge for the previous month (an update). However, this seems to have already been published and discussed in detail by Aguilar et al (2017), so what is the utility of the very brief repetition of the same work on a single river in this study? Given the results and methodology are far closer to Aguilar et al. (2017) than the rest of the submitted manuscript, it seems entirely out of place and only confirms their previous work.

We agree with the Reviewer that the application presented in Section 7 (which arguably is not *"incredibly short"*) whose theoretical basis is presented in section 2.3, is similar to what is presented by Aguilar et al. However, we refer here to a different river which has a higher memory with respect to the case studies previously analyzed and we also present a LFS application for the same river. Therefore we believe the case studied here is technically interesting. Sections 2.3 and 7 are titled "Technical experiment: Real-time updating of the frequency distribution of high and low flows" and "Real-time updating of the frequency distribution of high and low flows for the Oise River". They are meant to be a technical example. They do not present a scientific advance in the strict sense, but we believe they are an interesting addition to the paper. However, we may easily remove section 7 (and therefore section 2.3) if the Editor feels that they are redundant.

Our replies to the minor comments of the Reviewer follows here below.

**Figure 10 c, no colour scale provided**

We do not understand the comment as in our vision there is colour scale.

131: I don't understand the basis of correlating LFS with the mean flow of the previous month on the expectation this is a robust proxy for storage. If you define the LFS as the month with the lowest flow, then by definition the previous months will have higher flow, so how will this be a robust proxy for storage? In fact, you will be correlating against months that could also be included in the definition of HFS, which we would not suggest are a good indicator of storage.

Perhaps we missed the exact meaning of the comment, but in any case mass balance and energy balance apply to fluid mechanics and therefore river flow formation. Mass balance suggests that storage is related to river flow. The Reviewer, may refer to a simple conceptual model like the bucket model, where higher storage implies higher discharge and the river flow is clearly a proxy for storage. Besides, the Reviewer may feel free to use better proxies in his/her studies.

154: "SR (m3 s–1 km–2) is computed as the mean daily flow of the river standardized by the size of its basin area. It may be an important physical driver as it is an indicator of the catchment's wetness" – so this basically says that runoff can be considered as an indicator of how wet a catchment is. This is like saying rainfall can be considered an indicator of how much water is falling from the sky, hopefully the authors can see the silliness of such a statement without further explanation.

We are negatively surprised by the offensive tone used by the anonymous Reviewer. We do not see the reason why specific runoff should not be related to catchment wetness or aridity.

479: "This result was expected since the correlation refers to average flow while the HFS correlation is related to rapidly occurring events" See major points 2 -4, the design of the study is a major control on the results reported here rather than actual hydrological processes.

We regret to confirm that we fully disagree with the idea that an experiment should not be designed according to physical basis and scientific reasoning.

We respectfully submit a revised version of our paper. We regret to report that we do not agree with the criticism of the Reviewer and therefore did not make any major change to the manuscript in this respect, but only small clarifications (discussed above). We rely on the Editor assessment, in particular for the opportunity of keeping (or not) Section 7 and 2.3.

With our best regards,

Theano Iliopoulou, Cristina Aguilar, Berit Arheimer, María Bermúdez, Nejc Bezak, Andrea Ficchì, Demetris Koutsoyiannis, Juraj Parajka, María José Polo, Guillaume Thirel and Alberto Montanari

**1 A large sample analysis of European rivers on seasonal river flow correlation**

- 2 and its physical drivers
- 3 Theano Iliopoulou1\*, Cristina Aguilar2, Berit Arheimer3, María Bermúdez4, Nejc Bezak5, Andrea
- Ficchì6, Demetris Koutsoyiannis1, Juraj Parajka7, María José Polo2, Guillaume Thirel8 and
   Alberto Montanari9

[revised manuscript text omitted]

|           |                                                |                                                                                                                                         | $\mathbb{R}^2$                                                                                                                                                                                      |                                                                                                                                                                                              |
|-----------|------------------------------------------------|-----------------------------------------------------------------------------------------------------------------------------------------|-----------------------------------------------------------------------------------------------------------------------------------------------------------------------------------------------------|----------------------------------------------------------------------------------------------------------------------------------------------------------------------------------------------|
| 0.659407  | 0.008557                                       | 77.065                                                                                                                                  | < 2 ×10 -16 *** 0.58                                                                                                                                                                     | 334 104.2                                                                                                                                                                                    |
| -0.110632 | 0.006577                                       | -16.820                                                                                                                                 | $< 2 \times 10^{-16***}$                                                                                                                                                                            | p-value:                                                                                                                                                                                     |
| 0.031761  | 0.008070                                       | 3.936                                                                                                                                   | 0.000111***                                                                                                                                                                                         | $< 2.2 \times 10^{-16}$                                                                                                                                                                      |
| -0.038999 | 0.010388                                       | -3.754                                                                                                                                  | 0.000223***                                                                                                                                                                                         |                                                                                                                                                                                              |
|           | 0.659407
-0.110632
0.031761
-0.038999 | 0.659407         0.008557           -0.110632         0.006577           0.031761         0.008070           -0.038999         0.010388 | 0.659407         0.008557         77.065           -0.110632         0.006577         -16.820           0.031761         0.008070         3.936           -0.038999         0.010388         -3.754 | $R^{2}$ 0.659407 0.008557 77.065 <2 ×10 -16*** 0.58 -0.110632 0.006577 -16.820 <2 ×10 -16*** 0.031761 0.008070 3.936 0.000111*** -0.038999 0.010388 -3.754 0.000223*** |

**758 Figures**